# BREAKING ALGORITHMIC COLLUSION IN HUMAN-AI ECOSYSTEMS

## ABSTRACT

The growing adoption of AI agents is giving rise to ecosystems where these agents interact not only with each other but also with humans. We study such mixed ecosystems in the context of repeated pricing games, modeling AI agents as playing equilibrium strategies. We then analyze defections, where a human manually performs the task instead of their AI agent, thereby replacing an equilibrium player with a no-regret strategy. Motivated by how populations of AI agents can sustain supracompetitive prices, we ask whether high prices persist under such defections. Our main finding is that even a single human defection can destabilize collusion and drive down prices, while multiple defections push prices close to competitive levels. We further show how the nature of collusion changes under defection-aware AI agents and under greedy (follow-the-leader) defection strategies. Taken together, our results characterize when algorithmic collusion is fragile—and when it persists—in mixed ecosystems of AI agents and humans.

## 1 INTRODUCTION

As AI agents become capable of autonomously making decisions, humans are increasingly deploying AI agents to carry out tasks (e.g., pricing decisions, job applications, and content creation) on their behalf. The growing adoption of AI agents is giving rise to decentralized ecosystems where multiple deployed AI agents interact with each other at time (Rothschild et al., 2025), leading to rich emergent phenomena (Hammond et al., 2025).

When many companies in a market adopt AI agents for pricing decisions, this leads the pricing agents deployed by different companies to implicitly compete with each other over repeated interactions. A major concern is that these multi-agent interactions can drive *algorithmic collusion*. For example, RL agents can implicitly learn to collude with each other over repeated interactions: agents implement threat-based strategies where agents force each other to take specific actions by threatening retaliation at future time steps (Calvano et al., 2020a). Moreover, LLM agents can be steered towards collusion via prompting (Fish et al., 2025) and can secretly share proprietary information with each other (Motwani et al., 2025). When some humans opt to perform pricing decisions manually, these humans also interact with the AI agents at test-time. The resulting heterogeneity in these mixed ecosystems shapes the level of collusion that can be sustained (Werner, 2024; Schauer & Schnurr, 2023; Normann & Sternberg, 2022), raising the question of whether algorithmic collusion is robust to interactions between AI agents and humans.

In this work, we study algorithmic collusion from a theoretical perspective, building on the classical framework of *repeated Bertrand pricing games* (Fudenberg & Tirole, 1991). In these games, players repeatedly sell a good, and the player who offers the lowest price at each round wins and gains the profit of that sale. Collusion traces back to a fundamental property of repeated games: when players optimize for their long-run rewards, a vast set of pricing outcomes—including the high prices—can be realized at equilibrium. Motivated by recent empirical studies (Werner, 2024; Schauer & Schnurr, 2023; Normann & Sternberg, 2022), our work focuses on settings where some humans manually perform the pricing task rather than adopting an AI agent, which we refer to as a *defection*. In pricing contexts, defections could occur due to a lack of trust in algorithmic pricing tools (Bertini & Koenigsberg, 2021) or due to regulatory pressure to avoid collusion (Hartline et al., 2024). Such a defection would lead AI agents to compete not only with each other, but also with simple heuristic strategies deployed by humans. One might expect these defections to undercut high prices, eventually driving the prices back down to the competitive level and thus breaking collusion; however, this

intuition fails in 2-player pricing games, where high prices can be sustained even in the presence of simple heuristic strategies (Feldman et al., 2016; Arunachaleswaran et al., 2025).

We formally study when high prices persist under defections, building on the following stylized model of $N$-player pricing games. Borrowing from classical equilibrium concepts (Fudenberg & Tirole, 1991), we model AI agents as choosing strategies that constitute a Nash equilibria in the repeated game. This is motivated by how standard ML pipelines encourage agents to optimize for their long-run rewards, for example during RL training (Appendix B.1) or during LLM prompt specification (Appendix B.2). We model defections as following a generic no-regret strategy, capturing simple strategies such as greedy behavior with minor hedging that can approximate human behavior (Nekipelov et al., 2015).

Our main finding is that a single human defection can already drive the market price to be low, though not necessarily all the way down to the competitive price. More specifically, in any $N$-player algorithmic pricing equilibrium, if a "typical" player—one who attains at most the median profit in the system—defects, this drives the market price down by a polynomial factor in $N$. Moreover, if multiple humans defect, the market price falls exponentially in the number of defectors regardless of the identities of the players.

Our analysis tightly characterizes the market price under defections to no-regret strategies (Section 3). We show that any median-profit player defecting to a no-regret strategy always drives the time-averaged market price to at most $\frac{\ln(N)+1}{N}$ (Theorem 3.3). Furthermore, we construct an equilibrium which shows that this bound is asymptotically tight (Theorem 3.5). Together, these results illustrate that the stability of collusive high prices is inversely related to the number of competitors, making collusion harder to sustain in larger markets. When $M$ arbitrary players defect to no-regret strategies, we show that the market price reduces to $\frac{M}{e^{M-1}}$ (Theorem 3.6), and again show this bound is tight. At a high level, these reductions in market price stem from a core property of no-regret strategies in pricing games: they guarantee the player a share of the total market utility that is independent of the number of competitors. This guarantee, which is formalized in Lemma 3.4, fundamentally limits the ability of other agents to sustain collusive high prices.

To gain further insight into the market price in mixed human-AI ecosystems, we consider several model extensions (Section 4). First, we show that our analysis readily generalizes to a weaker equilibrium notion where players only need to prefer their strategy over a no-regret strategy, rather than over arbitrary strategies (Section 4.1). We then consider defection-aware AI agents who can dynamically adapt their strategies based on which humans defect. This defection-awareness fundamentally changes our results: the market price can remain high in the presence of the defection, regardless of the number of players $N$ (Section 4.2). Finally, we consider defections to a greedy strategy (FollowTheLeader). We show the market price after a defection can be as high as $\ln N/\sqrt{N}$, which establishes a separation between greedy defections and no-regret defections (Section 4.3)[1].

Taken together, our results demonstrate how algorithmic collusion can break down when even a very small number of humans defect to simple strategies. This provides a theoretical perspective on how emergent behaviors in human-AI ecosystems differ from those in ecosystems with only AI agents. More broadly, our work highlights how evaluators should thus account not only for interactions between AI agents, but also for interactions between these agents with humans.

## 1.1 RELATED WORK

**Algorithmic Collusion.** The potential for algorithms to learn collusive pricing strategies is an increasingly pressing concern. Early work demonstrated that RL agents can learn to sustain supracompetitive prices in simulated environments, typically by learning implicit punishment mechanisms (Calvano et al., 2020b; Klein, 2021). Collusion has also been empirically observed in real-world markets, where pricing algorithms are associated with elevated prices (Abada & Lambin, 2023; Assad et al., 2024; Wieting & Sapi, 2021). Recent work also highlights the possibility of LLM agents exhibiting secret collusion, in which coordination is intentionally concealed from monitors (Motwani et al., 2025). While the specific definition of collusion varies across works, a common property of many of

---

[1]When evaluating FTL defections, we consider a slightly different tiebreaking rule than in our standard model for technical simplicity. This is discussed in Section 4.3.

these definitions is that collusion leads to sustained high prices. Our analysis illustrates how any form of collusion satisfying this property is fragile to defections.

Prior work offers several possible explanations of algorithmic collusion. One explanation is that the equilibrium strategies emerging in a repeated game support collusion (Calvano et al., 2020a; Klein, 2021; Salcedo, 2015; Arunachaleswaran et al., 2025), for example via threats based on the opponent's past behaviors. Another explanation is that pricing algorithms take as input the opponents' current price but possess commitment power in a repeated game since they cannot be updated frequently (Brown & MacKay, 2023; Lamba & Zhuk, 2022; Salcedo, 2015). Other explanations include predictive alignment across algorithms due to statistical or model-driven linkages (Banchio et al., 2023; Jo et al., 2025) and coupling effects in RL and bandit pricing (Hansen et al., 2021). Our work builds on the perspective of equilibrium strategies emerging in a repeated game.

A handful of recent works study whether algorithmic collusion occurs when humans and AI agents interact with each other. Like our work, Normann & Sternberg (2022), Werner (2024) and Schauer & Schnurr (2023) also focus on repeated interactions, but these works empirically study collusion in laboratory experiments. Normann & Sternberg (2022) and Werner (2024) empirically demonstrate how increased adoption of AI agents leads to higher prices, and Schauer & Schnurr (2023) shows that prices in human–AI ecosystems can fall below those in all-human markets due to coordination challenges. Our work takes a step towards providing a theoretical foundation for these empirical observations. We note that Leisten (2024) also theoretically studies mixed ecosystems: however, while they study a pricing agent which chooses a price based on other players' prices (Brown & MacKay, 2023; Salcedo, 2015; Lamba & Zhuk, 2022) in a static game, we focus on pricing agents who chooses a price based on the history of past behaviors in a repeated game, which we believe better captures the realities of AI agents.

**No-regret dynamics in pricing games.** Our work builds on a growing line of work which investigates how no-regret dynamics interact with pricing games. Nadav & Piliouras (2010) show that, when *all* players employ no-regret strategies in a Bertrand pricing game, the resulting market price tends to the competitive price as the number of players increases. By contrast, we explore the interaction of no-regret agents with agents engaged in arbitrarily complex equilibrium strategies. Hartline et al. (2024) show that if all players employ no-swap-regret strategies, the market outcome is fully competitive, and propose auditing procedures for such guarantees. Follow-up work (Hartline et al., 2025) derives minimal properties of regret-minimizing strategies that suffice to guarantee competitiveness.

Our work also connects with learning dynamics in first-price auctions with a single item, which have an equivalence to Bertrand pricing games. For example, Kolumbus & Nisan (2022) analyze auctions between regret-minimizing agents, and Deng et al. (2022) and Feng et al. (2020) study dynamics of *mean-based* no-regret strategies. Closely related to our work, Feldman et al. (2016) characterize coarse correlated equilibria in single-item auctions, which correspond to the outcomes of no-regret dynamics. They show that the market price scales down exponentially in the number of players. Our analysis in Theorem 3.6 and Lemma 3.4 builds on proof techniques in Feldman et al. (2016).

Closely related to our work, Arunachaleswaran et al. (2025) construct two-player equilibria with a no-regret strategy that sustain high prices and show that mean-based no-regret dynamics converge to competitive prices. A key difference is that our model focuses on defection to no-regret strategies rather than an equilibrium with a no-regret strategy, and we prove upper bounds in this defection model. While the analysis in Arunachaleswaran et al. (2025) implies a lower bound on the market price under defections for $N = 2$ players, our equilibrium constructions improve upon this bound, and also generalize to $N \geq 2$ players. Our analysis in Lemma 3.4 builds on proof techniques in Arunachaleswaran et al. (2025).

## 2 MODEL

We formalize the repeated game in Section 2.1, player strategies in Section 2.2, and the market price in Section 2.3. We discuss how our model captures key aspects of pricing ecosystems with RL agents and with LLM agents in Appendix B, and we discuss model limitations in Section 5.

### 2.1 REPEATED PRICING GAME

We consider a market based on the repeated pricing games (Fudenberg & Tirole, 1991).

In the standard one-shot (or static) version of this game, $N$ players (sellers) simultaneously choose a (possibly randomized) price over a discretized set $\{0, 1/K, \ldots, 1\}$. The payoffs in this game capture how the seller who offers the lowest price attracts all of the customer demand. More specifically, the payoffs are specified by the standard Bertrand rule, where the seller who selected the lowest price gets payoff equivalent to that price, and all other players get no payoff. In the case of ties, the payoff is divided evenly. We assume that the cost is zero.

We study the *repeated* version of this game, played over $T$ rounds. In each round $t \in [T]$, every player $i \in [N]$ chooses a *(possibly randomized) price* $P_{i,t}$ over the price set. At the end of each round, a price $p_{i,t} \sim P_{i,t}$ is drawn for each player, and payoffs $u_{i,t}(p_{i,t}; p_{-i,t})$ are awarded according to the Bertrand rule. Formally:

- If exactly one player $j$ chooses the strictly lowest price $p = \min_{i \in [N]} p_{i,t}$, then the payoff of this player is $u_{j,t}(p_{j,t}; p_{-j,t}) = p$, and every other player $i \neq j$ gets payoff $u_{i,t}(p_{i,t}; p_{-i,t}) = 0$.
- If $I \subseteq [N]$ players are tied for the lowest price $p = \min_{i \in [N]} p_{i,t}$, each player $i \in I$ obtains a payoff of $u_{i,t}(p_{i,t}; p_{-i,t}) = \frac{p}{|I|}$, while all other players $i \notin I$ receive payoff $u_{i,t}(p_{i,t}; p_{-i,t}) = 0$.

We will slightly abuse notation and let $u_{i,t}(P_{i,t}, P_{-i,t}) = \mathbb{E}[u_{i,t}(p_{i,t}, p_{-i,t})]$ denote the expected payoff, where the expectation is over the realized prices $p_{j,t} \sim P_{j,t}$ for each player $j \in [N]$. We also let $\lceil p \rceil_K$ (resp. $\lfloor p \rfloor_K$) denote the rounding of $p$ to the closest lower (resp. higher) discretization of $\frac{1}{K}$.

## 2.2 Repeated Game Strategies and Equilibrium

We model player strategies as mapping histories to price distributions, which captures how players can adapt to the surrounding environment but fix their strategy ahead of time. The equilibrium concept is Nash equilibrium in strategy space, which captures an equilibrium in the repeated game (Fudenberg & Tirole, 1991). We model defection (i.e., humans performing the task manually) as the player switching to a no-regret strategy.

**Strategies.** Each player $i$ has a repeated-game *strategy* $S_i$ that determines a distribution over prices $P_{i,t}$ at every round $t$ based on the history of prices and payoffs of all players from round $1$ to $t - 1$. Let $\mathcal{S}_{1:N} = (S_1, \ldots, S_N)$ be the strategy profile of all players' strategies. Let $U_i(\mathcal{S}_{1:N}) = \frac{1}{T} \sum_{t=1}^{T} u_{i,t}(P_{i,t}, P_{-i,t})$ denote the expected average utility of player $i$ across all time steps $T$.[2]

**Equilibrium concept.** We focus on $\epsilon$-approximate equilibria. More formally, an $\epsilon$-approximate *repeated game equilibrium* is an approximate Nash equilibrium in strategy space: a strategy profile $\mathcal{S}^* = (S_1^*, \ldots, S_N^*)$ is a repeated game equilibrium in a $T$-round game if, for all players $i$, it holds that $U_i(S_i^*, S_{-i}^*) \geq \max_{S_i \in \mathcal{S}(T)} U_i(S_i, S_{-i}^*) - \epsilon$. In other words, no player improves their expected utility by more than $\epsilon$ if they swap out their strategy for a different one, keeping the strategies of all other players fixed. We will typically take $\epsilon = \Theta(1/T)$. The set of $\Theta(1/T)$-approximate equilibria is very rich and can range anywhere from the lowest price of $0$ to the highest price of $1$ (Fudenberg & Tirole, 1991). We discuss our rationale for focusing on approximate equilibria in Appendix D.

**Defections.** We will primarily focus on defection to a *No-Regret (NR) Strategy* $S^r$, motivated by behavioral assumptions in prior work (Nekipelov et al., 2015). A strategy $S^r$ has $r(T)$ regret for player $i$ if $\left( \max_{p \in \{0, 1/K, \ldots, 1\}} \sum_{t=1}^{T} u_i(p, P_{-i,t}) \right) - \left( \sum_{t=1}^{T} u_i(P_{i,t}, P_{-i,t}) \right) \leq r(T)$ where $P_{i,t}$ is the output of $S^r$ and where the $P_{-i,t}$ come from any (consistent) sequence of histories. $S^r$ is *no-regret* if $r(T) = o(T)$. We extend our analysis to greedy defections in Section 4.3.

We use the word defection to refer to an agent changing from their equilibrium strategy to a no-regret strategy (or, in Section 4.3, to an FTL strategy). Defections represent a human performing a pricing task manually in place of their AI agent. It will sometimes be useful to discuss other types of strategies that an agent may switch to from equilibria; we refer to these as more general "deviations".

## 2.3 Market Price

The *market price* induced by the players' strategies captures the market's level of uncompetitiveness. Given a strategy profile $\mathcal{S}_{1:N}$, the *market price*

$$\text{Price}(\mathcal{S}_{1:N}) = \frac{1}{T} \sum_{t=1}^{T} \min_{i \in [N]} (p_{i,t})$$

---

[2]We formalize the history in Appendix D.

captures the average price set by the winning player at each round. We use a price of 0 as a benchmark for a perfectly competitive market, since any Nash equilibria in the static one-round game leads to a market price $\leq 2/K$ for sufficiently large $K$, which approaches 0 as $K \to \infty$. Since the market price equals the expected cumulative utility of the winning players[3], the market price also captures how much profit is transferred to the players selling the good from consumers purchasing the good.

**Market price under defections.** Our focus is on how the market price changes under defections. Specifically, suppose that all $N$ players initially choose an $\epsilon$-approximate equilibrium strategy profile $\mathcal{S}^* = (S_1^*, \ldots, S_N^*)$, but player $i$ *defects* by switching to a strategy $S_i \in \mathcal{S}(T)$. We study the resulting market price

$$\text{DefectedPrice}(i, \mathcal{S}^*, S_i) := \text{Price}(S_1^*, \ldots, S_{i-1}^*, S_i, S_{i+1}^*, \ldots, S_N^*)$$

under this defection. We will also consider defections by a set of players $I \subseteq [N]$ to strategies $S_i$ for $i \in I$, and define the defected price $\text{DefectedPrice}(I, \mathcal{S}^*, \{S_i\}_{i \in I})$ analogously.

## 3 Main Results: Defections to no-regret strategies

We characterize how no-regret defections impact the market price, and show that the impact is tightly connected to the number of agents in the market $N$. First, as a warmup, we construct a simple equilibrium profile that shows how after a no-regret defection, the market price can remain as high as $1/N$ (Section 3.1). We then show that no matter what the initial equilibria is, a no-regret defection by a median-profit player always drives the market price to be low in large markets: the market price is upper bounded by $(1 + \ln N)/N$ (Section 3.2). To show this bound this tight, we modify the warmup construction so that the market price is on the order of $(1 + \ln N)/N$ after a defection (Section 3.3). Finally, we generalize these results to defections by $M$ arbitrary players at once (Section 3.4).

### 3.1 Warmup

A single no-regret defection can significantly drive down the equilibrium price. To gain intuition for this, consider the following repeated game equilibrium which supports collusion via *threats*. Specifically, consider the $\frac{1}{T}$-approximate equilibrium where all players use the following strategy: choose 1 as long as nobody has ever priced below 1, and choose price 0 otherwise. The market price at this equilibrium is 1. While all players set the price to 1 at every round at equilibrium, a no-regret defection will quickly begin to undercut the price of 1. This will trigger the threats of the other strategies and plummet the market price to 0.

This collapse at a single defection may seem unavoidable in any high-price approximate equilibrium. However, this intuition already fails in the two-player case (Arunachaleswaran et al., 2025). As we show below, the intuition also fails in $N$ player games.

To illustrate this in our defection model, we construct a simple (but suboptimal) approximate equilibrium where a no-regret defection by any player results in a market price of $\approx \frac{1}{N}$. The key is orchestrating threats which are enough of a deterrent to maintain the equilibrium, but which still induce a reasonably high price against defections. Let $\mathcal{S}^{\text{simple}} = [S_1^{\text{simple}}, \ldots, S_N^{\text{simple}}]$ be the strategy profile where every player uses the following strategy: choose a price of 1 as long as nobody has ever priced below 1 and choose a price $\lfloor \frac{1}{N} \rfloor_K$ otherwise. We show that $\mathcal{S}^{\text{simple}}$ is an approximate equilibrium and results in a market prices of approximately $1/N$ under a no-regret defection.

**Proposition 3.1** (Warm-up). *Let $N \geq 2$, $K > 1$, $T \geq 1$. Let $\mathcal{S}^{\text{simple}}$ be defined as above, and for $i \in [N]$, let $S_i^r$ be any $r(T)$-regret strategy for player $i$. Then, $\mathcal{S}^{\text{simple}}$ is an $O(1/T)$-approximate repeated game equilibrium with a market price of $\text{Price}(\mathcal{S}^{\text{simple}}) = 1$. Moreover, for any player $i \in [N]$, it holds that:*

$$\text{DefectedPrice}(i, \mathcal{S}^{\text{simple}}, S_i^r) \geq \frac{1}{N} - \frac{2}{K} - \frac{r(T)}{T}.$$

Proposition 3.1 shows that in the limit as $K \to \infty$ and $T \to \infty$, the market price can be as high as $1/N$ when a player defects to any no-regret strategy. The intuition is that the strategy profile $\mathcal{S}^*$ captures a threat of switching to $\approx 1/N$ prices if any player deviates from setting the price to be 1.

---

[3] That is, $P(\mathcal{S}_{1:N}) = \sum_{i \in I} U_i(\mathcal{S}_{1:N})$, where $I = \arg\min_{i \in [N]} p_{i,t} \subseteq [N]$ players is set of players tied for the lowest price.

While this threat is weaker than the "price at 0" threat, it is sufficient to disincentivize players from undercutting. And when a player defects to a no-regret strategy and triggers the threat, the price is now substantially above 0—the defecting player will learn to set their price right below $1/N$ and always win. The proof formalizes this intuition and is deferred to Appendix E.1.

## 3.2 Upper Bound: Low Market Price in Large Markets

While Proposition 3.1 illustrates that nonzero prices can be sustained under no-regret defections, we show that the defection still drives down the market price in large markets. Specifically, at any equilibrium, the market price scales down with the number of players in the market under a median-profit assumptions on the player who chooses the no-regret defection (Theorem 3.3).

To illustrate why such an assumption is necessary, we construct an approximate equilibrium where defection by the highest-profit player leads to a market price close to 1, regardless of the number of other players in the market.

**Proposition 3.2.** *Let $N \geq 3$, $K > 1$, and $T \geq 1$. For each player $i \in [N]$, let $S_i^r$ be any $r(T)$-regret strategy for player $i$. There exists a $O(1/T)$-approximate repeated game equilibrium $\mathcal{S}^*$ with a market price of $Price(\mathcal{S}^*) = 1$ where:*

$$DefectedPrice(i_N(\mathcal{S}^*), \mathcal{S}^*, S_i^r) \geq 1 - \frac{1}{K} - \frac{r(T)}{T},$$

*where $i_N(\mathcal{S}^*) \in [N]$ is the highest-profit player.*

The intuition for Proposition 3.2 is that there is a pathological approximate equilibrium where the players $i \neq i_N(\mathcal{S}^*)$ place stronger threats on each other than they place on $i_N(\mathcal{S}^*)$. These stronger threats significantly constrain of the behavior of these players: at this approximate equilibrium, players $i \neq i_N(\mathcal{S}^*)$ choose a price of 1, even though player $i_N(\mathcal{S}^*)$ chooses a price of $1 - 1/K$. If player $i_N(\mathcal{S}^*)$ defects to a no-regret strategy, then the market price continues to be high.

To avoid pathological equilibria designed around a specific player, we consider defections by a "typical player", which we formalize as a player who achieves at most the profit of the median player. Intuitively, this assumption only excludes high-profit players who likely have market power. We formalize the median-profit set as follows: Given an equilibrium $\mathcal{S}^* = (S_1^*, \ldots, S_N^*)$, we order the players in $[N]$ in increasing order of profit, so that $i_j(\mathcal{S}^*)$ is the identity of the player with the $j$th lowest profit: that is, $U_{i_1(\mathcal{S}^*)}(S_{i_1(\mathcal{S}^*)}^*, S_{-i_1(\mathcal{S}^*)}^*) \leq U_{i_2(\mathcal{S}^*)}(S_{i_2(\mathcal{S}^*)}^*, S_{-i_2(\mathcal{S}^*)}^*) \leq \ldots \leq U_{i_N(\mathcal{S}^*)}(S_{i_N(\mathcal{S}^*)}^*, S_{-i_N(\mathcal{S}^*)}^*)$. We define *median profit set* $\mathrm{I}^{\mathrm{AP}}(\mathcal{S}^*)$ to be $\left\{ i_j(\mathcal{S}^*) \mid j \leq \lfloor \frac{N}{2} \rfloor \right\}$.

When we restrict to defections by median-profit players $i \in \mathrm{I}^{\mathrm{AP}}(\mathcal{S}^*)$, we obtain the following upper bound.

**Theorem 3.3.** *Let $N \geq 2$, $K > 1$, and $T \geq 1$. For each player $i \in [N]$, let $S_i^r$ be any $r(T)$-regret strategy for player $i$. Let $\mathcal{S}^* = (S_1^*, \ldots, S_N^*)$ be any $\epsilon$-approximate repeated game equilibrium, where $\epsilon = O(1/T)$. For any player $i \in \mathrm{I}^{\mathrm{AP}}(\mathcal{S}^*)$, it holds that*

$$DefectedPrice(i, \mathcal{S}^*, S_i^r) = O\left( \frac{\ln N + 1}{N} + (\frac{r(T) + 1}{T}) \ln(N) + \frac{1}{K} + \frac{1}{T} \right).$$

Theorem 3.3 demonstrates that a single player defecting to a no-regret strategy can already drive down the market price when there are sufficiently many other players in the market. However, the price reduction does rely on the defecting agent being in the median-price set, since Proposition 3.2 demonstrates higher prices are achievable if the highest-profit player defects instead. This suggests, perhaps counterintuitively, that regulators auditing for collusion in a market ecosystem may benefit from auditing and enforcing no-(swap-)regret behavior in the lower-profit players, rather than solely focusing on high-profit players.

**Proof ideas.** To prove Theorem 3.3, the key technical ingredient is to upper bound the market price in terms of the utility of the defecting agent who uses a no-regret strategy (Lemma 3.4).

**Lemma 3.4.** *Let $N \geq 2$, $K > 1$, and $T \geq 1$. Let $i \in [N]$ be any specific player, and suppose that $S_i^r$ is a $r(T)$-regret strategy for player $i$. If $U_i(S_i, S_{-i}) \leq c$, then the market price is at most*

$$c + \frac{1}{K} + \left( c + \frac{r(T)}{T} \right) \left( 1 + \ln \left( \frac{1}{c} \right) \right).$$

The proof Lemma 3.4 builds on ideas from Arunachaleswaran et al. (2025) and Feldman et al. (2016), and is deferred to Appendix E.4.

### 3.3 Optimal construction for a no-regret defector

We now construct an equilibrium that achieves a market price of $\approx (\ln N + 1)/N$ for large $K$ and $T$, improving upon the bound in Proposition 3.1.

**Theorem 3.5.** *Let $N \geq 2$, $K > 1$, and $T \geq 1$. For each $i \in [N]$, let $S_i^r$ be any $r(T)$-regret strategy for player $i$. There exists an $O(1/T)$-approximate repeated game equilibrium strategy profile $\mathcal{S}^*$ with market price $Price(\mathcal{S}^*) = 1$ such that for any player $i \in [N]$, it holds that:*

$$DefectedPrice(i, \mathcal{S}^*, S_i^r) = \frac{\ln(N) + 1}{N} - \frac{3\ln(N)}{K} - O\left(\sqrt{\frac{r(T)\ln(N+1)}{T}}\right).$$

**Proof ideas.** The construction in Theorem 3.5 leverages two key ideas, which we summarize below. We defer the full proof to Appendix E.2.

The first idea is to replace the threat of $\lfloor 1/N \rfloor_K$ in Proposition 3.1 with a carefully tuned *randomized* price $P$. This randomized price distribution is a discretization of an *equal revenue distribution* from auction theory . The equal revenue distribution $P_c$ with parameter $c \in (0, 1)$ is supported on $(c, 1]$ and has the property that in a one-round pricing game with 2 players where one player chooses $P_c$, the other player achieves their optimal payoff $c$ by choosing any price $p \in (c, 1)$. We take $P = P_{1/N,K,\epsilon}^{\text{high}}$ to be a discretized version of the equal revenue distribution with parameter $c = 1/N$ where the distribution is also $\epsilon$-perturbed to make the second-highest price $1 - 1/K$ yield a slighter higher revenue than the other prices.

The intuition for why the threat-distribution $P = P_{1/N,K,\epsilon}^{\text{high}}$ leads to high prices can be seen in two-player games. First, both players choosing the strategy of pricing at 1 using $P$ as a threat is an (approximate) equilibrium, due to the revenue properties of $P_{1/N,K,\epsilon}^{\text{high}}$. Moreover, the market price is higher than $1/N$, since the no-regret defector chooses a price $1 - 1/K$ in almost all of the rounds.

The second idea enables us to generalize this intuition to $N$ players. At first glance, it may be tempting to have all $N - 1$ players implement the threat distribution $P$ upon any deviation from the price of 1. However, this leads the market price to be too low, since the market affected is affected by the minimum of $N - 1$ random samples from this distribution. To avoid this, we modify the structure of threats. We introduce *cyclic threats* where only the player $(i + 1) \mod N$ triggers a threat when player $i$ defects.

**Implications of Theorem 3.3 and Theorem 3.5.** The bound in Theorem 3.5 as $K \to \infty$ and $T \to \infty$ achieves matches the bound in Theorem 3.3. This means that Theorem 3.3 and Theorem 3.5 together characterize the market price when players in the median-price set defect to no-regret strategies: the bound in Theorem 3.5 and the bound in Theorem 3.3 both equal $\Theta((\ln +1)/N)$ as $K \to \infty$ and $T \to \infty$. This bound asymptotically characterizes the market price, and illustrates how the market price scales approximately inversely with the number of players $N$.[4]

Theorem 3.5 improves upon the bounds implied by prior work. Specifically, the analysis shown in Arunachaleswaran et al. (2025) implies that in a two-player game, the market price under a defection can be as large as $2/e$. Note that for 2 players, the construction in Theorem 3.5 achieves an improved bound of $\approx (1 + \ln 2)/2 > 2/e$. Theorem 3.5 also generalizes this bound to $N \geq 2$ players.

### 3.4 Low Market Prices under Multiple No-regret Defections

We next turn to the case of *multiple* no-regret defections and also relax the assumption that the other agents start at an approximate equilibrium. We show that the market price exponentially reduces with the number of defectors, even if the other agents' strategies are not at equilibrium.

**Theorem 3.6.** *Let $N \geq 2$, $K > 1$, and $T \geq 1$. Let $S$ be any strategy profile, not necessarily in equilibrium. Let $I \subseteq [N]$, let $M := |I|$, and let $\{S_i^r\}_{i \in I}$ be such that $S_i^r$ is an $r(T)$-regret strategy*

---

[4]Note that if we had strengthened the median-profit assumption on the defecting player to a stronger assumption that the *lowest*-profit player defects, then our upper bound to be exactly tight in terms of $N$.

*for each $i \in [N]$. If $r(T) = o(T)$, then it holds that:*

$$DefectedPrice\left(I, S, \{S_i^r\}_{i \in I}\right) \leq \frac{M}{e^{M-1}} \cdot (1 + o_T(1) + o_K(1))$$

We complement Theorem 3.6 with a matching upper bound where players start at an equilibrium, which we defer to Appendix E.7. This, together with Theorem 3.6, shows a tight bound of $\Theta\left(\frac{M}{e^{M-1}}\right)$ on the worst-case market price with $M$ no-regret defectors when the other agents play arbitrary strategies. This demonstrates that the presence of sufficiently many agents defecting to no-regret strategies guarantees exponentially low prices.[5]

To prove these results, we build on the proof techniques in Feldman et al. (2016). Specifically, the results in Feldman et al. (2016) imply that when $K \to \infty$, the maximum possible market price at a coarse correlated equilibrium is $M/e^{M-1}$. Given the correspondence between no-regret strategies and coarse correlated equilibria, this immediately implies that the maximum possible price in a game with $M$ no-regret strategies is $M/e^{M-1}$. We defer the proof of Theorem 3.6 to Appendix E.6.

## 4 EXTENSIONS

In this section, we consider model extensions to a weaker notion of equilibrium (Section 4.1), to agents that can adapt their strategies at test-time (Section 4.2), and to greedy defections (Section 4.3).

### 4.1 ADOPTION EQUILIBRIUM

We relax the assumption that player strategies form a repeated-game equilibrium. Specifically, suppose that each player is only guaranteed to be incentivized to choose $S_i^*$ over a no-regret defection: i.e., for each player $i \in [N]$, it must hold that $U_i(S_i^*, S_{-i}^*) \geq U_i(S_i, S_{-i}^*) - \epsilon$ for any no-regret strategy $S_i$, but not necessarily other strategies. We call this equilibrium concept an *adoption equilibrium*. At a high level, a repeated game equilibria captures AI agents being trained to play against each other (Appendix B.1), whereas the adoption equilibrium only guarantees that humans prefer to adopt the AI agent over setting prices manually. Interestingly, Theorem 3.3 and Theorem 3.5 directly generalize to adoption equilibria, since the only fact that we used about equilibria is that the players prefer their equilibrium strategy to their payoff from a no-regret defection.

### 4.2 DEFECTION-AWARE AI AGENTS

While we have thus far assumed that AI agents are committed to pre-existing equilibrium strategies, we now allow AI agents to adapt their strategies at test-time, fully aware of the human's presence. We envision that this test-time awareness may arise as AI agents become more capable of dynamically shifting their long-run policies in response to changes in the test-time environment (Appendix B.1 and Appendix B.2). To model this, we modify the equilibrium assumption for the AI agents: now, we assume that after defections take place, the AI agents form a new equilibrium with awareness of which players defected and what strategies they chose. We formalize this in Definition F.1.

When the AI agents can form their equilibria given the context of the human's defection, *high market prices can be sustained even in the presence of a no-regret defection*. The following theorems establish that the market price can be close to 1 regardless of the number of players $N$. The key distinction is that defection-aware AI agents are not constrained to play in such a way that guarantees the no-regret defector a low utility, as they are not tied to playing the off-path sequence of an equilibrium.

**Theorem 4.1.** *Let $N \geq 2$, $K > 1$, and $T \geq 1$. Fix a designated player $i \in [N]$ who runs a strategy $S_i^r$ with regret $r(T)$. Then there exists an $O(1/T)$–approximate defection-aware repeated-game equilibrium $\mathcal{S}_{-i}$ with respect to $(i, S_i^r)$ (Definition F.1) such that*

$$Price\left(\mathcal{S}_{-i}, S_i^r\right) \;\geq\; 1 - \frac{1}{K} - \frac{r(T)}{T}.$$

Theorem 4.1 gives asymptotically tight upper and lower bounds for the highest possible price, as the highest price is trivially 1. We analyze how welfare is distributed across players in Appendix F.2.

---

[5]Since Theorem 3.6 does not require a median-profit assumption, it is weaker than the bound in Theorem 3.3 in the case of $M = 1$. Nonetheless, as long as $M = \Omega(\log N)$, the bound is smaller than that of Theorem 3.3.

### 4.3 Follow the Leader Defections

Our final extension considers a different type of defection: the Follow-the-Leader (FTL) strategy. Unlike a no-regret strategy, which optimizes its own historical performance, FTL is purely reactive and has an explicit undercutting flavor. The FTL strategy $S_i^{\text{FTL}}$ chooses a *deterministic* price $P_{i,t}$ that is the best-response to the empirical distribution of prices played by the other players: that is, $S_i^{\text{FTL}}(H_{t-1}) \in \arg\max_{p \in \{0, 1/K, \ldots, 1\}} \frac{1}{t-1} \sum_{t'=1}^{t-1} u_i(p, P_{-i,t})$.

We show *a separation between the market price achievable under no-regret defections and FTL defections*. The following result shows that an FTL defection can result in a strictly higher market price that exceeds the bound for no-regret defections in Theorem 3.3. To simplify the behavior of FTL, we model the FTL player as winning tiebreaks in this section. This is in contrast to the rest of the paper, where we model tiebreaking as uniform.

**Theorem 4.2.** *Let $N \geq 2$, $K > 1$, $T \geq 1$. There exists an $O(1/T)$-approximate repeated game equilibrium strategy profile $\mathcal{S}^*$ with a market price of $\text{Price}(\mathcal{S}^*) = 1$ such that for any player $i \in [N]$, it holds that*
$$\text{DefectedPrice}(i, \mathcal{S}^*, S_i^{FTL}) = \Omega\left(\frac{\ln N}{\sqrt{N}}\right).$$

The construction for Theorem 4.2 involves a choreographed punishment mechanism similar to the one in Section 3.3. When a player $i$ defects to FTL, a designated "punisher" player $\pi(i)$ switches to a stochastic pricing strategy based on a cycle of distributions. This cycle is designed to exploit the FTL player's reactive nature, leading it to choose prices that result in a market price of $\Omega(\frac{\ln N}{\sqrt{N}})$. The proof, which details the construction of this cycle, is deferred to Appendix G.1.

## 5 Discussion

In this work, we investigate algorithmic collusion in pricing ecosystems, where some humans delegate their pricing decisions to AI agents and others opt to manually perform the pricing task themselves. To formalize this, we study a repeated Bertrand pricing game where we model AI agents as choosing strategies forming a Nash equilibrium in the repeated game. We model human defections as no-regret strategies. Our main finding is that a single human defection drives the price to $\Theta(\ln N/N)$, as long as the human originally obtains at most the median profit, and that this bound is tight. We also characterize the market price under $M$ defectors, greedy defectors, and defection-aware AI agents.

At a conceptual level, our work offers a theoretical perspective on how algorithmic collusion in human-AI pricing ecosystems is fundamentally different than in ecosystems with only AI agents. Our results highlight the fragility of algorithmic collusion to even a single human not adopting the AI agent, providing theoretical support for empirical observations (Werner, 2024; Schauer & Schnurr, 2023; Normann & Sternberg, 2022). This suggests that multi-agent evaluations with only AI agents can fail to capture emergent behaviors in real-world ecosystems, which rarely exhibit full adoption, and highlights the importance of accounting for human agents in evaluations (Carroll et al., 2020).

**Limitations and Future Work.** Our stylized model makes several simplifying assumptions for analytic tractability, and we view relaxing these assumptions as an important direction for future work. For example, we assume that AI agents play equilibrium strategies in the repeated game, motivated by how standard ML pipelines encourage agents to optimize for long-run objectives during RL training (Appendix B.1) or LLM prompt specification (Appendix B.2). We showed that our results generalize readily to the the weaker notion of adoption-aware equilibria (Section 4.1), but our model still abstracts away many of the details of model training. It would be interesting to specialize our model to a specific type of AI agent and more closely capture the strategies learned by that agent. Moreover, we assume that defections take the form of no-regret strategies or greedy strategies. It would be interesting to investigate the robustness of our results when this class of strategies is broadened for example to gradient-based updates or state-based strategies. Finally, our pricing game assumes homogeneous goods and non-noisy consumer decisions; it would be interesting to incorporate smoother demand curves as well as heterogeneous goods and costs.

## 6 Reproducibility Statement

All claims made in the body of this paper have complete proofs in the Appendix.

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

## A  LLM USAGE STATEMENT

LLM USAGE STATEMENT We used GPT-5, Gemini 2.5 within the Windsurf environment, and Claude Opus 4.1 as aids in editing prose and searching for related work. Any language modified by an AI model or citations found by an AI model were carefully evaluated by the human authors.

## B  REAL-WORLD EXAMPLES

We discuss how our model captures key aspects of pricing ecosystems with RL agents as well as pricing ecosystems with LLM agents. In both examples, we model defections as humans playing no-regret strategies, motivated by how this behavioral assumption is used in auctions (Nekipelov et al., 2015).

### B.1  CONNECTION TO RL AGENTS

Consider an ecosystem where companies deploy RL agents to make pricing decisions on their behalf (Calvano et al., 2020a).

The strategy $S$ in our model captures the policy that the RL agent learns during the training. An RL agent's policy produces different outputs depending on the world state, which is captured by the output of the strategy $S$ being adaptive to the actions taken by other players. We envision that the RL agent is trained to perform well under multi-agent interactions, for example via self-play (Silver et al., 2017).

Our model takes an idealized and simplified view of these training dynamics: we assume that RL agents implement a Nash equilibria in the repeated game, motivated by how training approaches based on self-play converge to a Nash equilibrium in idealized finite two-player zero-sum games. An RL agent's policy is typically fixed after training, which is captured by the player being unable to change its strategy $S$ at test-time even if other players perform defections.

We note that some RL training approaches do account for human behavior during training (Carroll et al., 2020; Meta Fundamental AI Research Diplomacy Team, 2022); we view these RL agents as exhibiting greater robustness to human defections, though these agents are still unlikely to exhibit full defection-awareness (Section 4.2) since the policy is still fixed at test-time.

### B.2  CONNECTION TO LLM AGENTS

Suppose that companies deploy LLM agents to make pricing decisions on their behalf. These companies can influence the LLM agent's behavior through prompt engineering. We view the strategy $S$ in our model as capturing the "policy" that the LLM agent implements for a given choice of prompt. Here, the policy captures the output of the LLM agent in response to the history of prior play (which we envision is captured in its context).

Recent work suggests that when LLMs are prompted to optimize for a long-run objective, the interactions between LLMs can lead to high prices in pricing games (Fish et al., 2025), and that these agents can increasingly implement long-run strategic reasoning in repeated games (Zhang et al., 2024). An LLM agent's policy is fixed at test-time, which is captured by the player being unable to change its strategy in response to defections. However, as LLMs become increasingly capable, it is possible that these agents will implement policies that detect and adapt to defections, as captured by the defection-aware model in Section 4.2.

## C  PRELIMINARIES ON EQUAL REVENUE DISTRIBUTIONS

Many of our proofs rely on constructions using the Discretized Equal Revenue Distribution (ERD), which we formally define here. This is a discretization version of the continuous equal-revenue distribution. The continuous ERD with parameter $c \in (0, 1)$ is supported on $[c, 1)$ and has cumulative distribution function given by:

$$\begin{cases} 0 & \text{if } x < c \\ 1 - \frac{c}{x} & \text{if } x \in [c, 1) \\ 1 & \text{if } x = 1. \end{cases}$$

The key property of this distribution, which is inherited by the discretized ERD with parameter $c$, is that it equalizes the expected revenue for any price $p \geq c$. The expected value of a price drawn from an ERD is closely related to the harmonic series and is on the order of $\Omega(c \log(1/c))$.

**Definition C.1** (Discretized Equal Revenue Distribution (DERD) $P_{c,K}$). *Let $c = m/K$ for some integer $m \in \{1, \ldots, K-1\}$. The discretized equal-revenue distribution (ERD) $P_{c,K}$ over the price grid $\{0, 1/K, \ldots, 1\}$ is defined by the probability mass function:*

$$P_{c,K}(p_i) = \begin{cases} c & \text{if } i = K \\ c\left(\frac{1}{p_i} - \frac{1}{p_{i+1}}\right) & \text{if } i \in \{m, \ldots, K-1\} \\ 0 & \text{otherwise} \end{cases}$$

*where $p_i = i/K$. This distribution ensures that the expected utility $\mathbb{E}_{X \sim P_{c,K}(p_i)}[p \cdot 1[X \geq p]]$, where ties are broken in favor of this player, is exactly $c$ for any price $p \geq c$.*

*Remark* 1. $P_{c,K}$ does not guarantee utility exactly $c$ in our model, as by contrast to the tiebreaking assumption above, we employ standard uniform tiebreaking. However, the utility difference is on the order of $\frac{1}{K}$ for all prices $< 1$ and is absorbed into our $\frac{1}{K}$ error terms. In Section 4.3, we do break ties in favor of the player playing against the DERD.

**Harmonic Numbers and DERD Expectation.** For the discretized equal revenue distribution $P_{c,K}$ supported on grid points $\{m/K, \ldots, 1\}$ with $m = \lceil cK \rceil$, the exact expectation can be expressed in terms of harmonic numbers:

$$\mathbb{E}[P_{c,K}] = c(H_K - H_m + 1),$$

where $H_n = \sum_{j=1}^{n} 1/j$. Using the standard bounds $\ln \frac{K+1}{m+1} \leq H_K - H_m \leq \ln \frac{K}{m}$, we obtain

$$c\left(\ln \frac{K+1}{m+1} + 1\right) \leq \mathbb{E}[P_{c,K}] \leq c\left(\ln \frac{K}{m} + 1\right).$$

In particular, when $m = cK$, this shows $\mathbb{E}[P_{c,K}] = c(\ln(1/c) + 1) \pm O(1/K)$, matching the continuous ERD expectation to within $\frac{1}{K}$.

# D   OMITTED DETAILS FROM SECTION 2

**Formalization of history.** More formally, let the history $H_t = \{P_{i,t'} \mid i \in [N], 1 \leq t' \leq t\} \cup \{u_i(p, P_{-i,t'}) \mid i \in [N], 1 \leq t' \leq t, p \in \{0, 1/K, \ldots, 1\}\}$ denote the price distributions and expected payoffs up to and including the time step $t$, and let $\mathcal{H}$ be the set of all possible histories at any time step. A strategy $S : \mathcal{H} \to \Delta(\{0, 1/K, \ldots, 1\})$ maps histories to price distributions. Let $\mathcal{S}(T)$ denote the set of all possible strategies operating over $T$ rounds. Let $S_i \in \mathcal{S}(T)$ be player's $i$'s strategy, which means that they choose price distribution $P_{i,t} = S_i(H_{t-1})$ at each round $t$.

**Rationale for considering approximate equilibria.** Our rationale for considering approximate equilibria is to make the model faithful to the strategic phenomena we wish to capture. In finitely repeated games, the set of exact equilibria is artificially narrow: by backward induction, collusion collapses even in settings where empirical evidence and simulations show that high prices can persist. By instead focusing on $\Theta(1/T)$-approximate equilibria, we capture the natural regime for long-horizon learning: when $T$ is large, small deviations are negligible relative to long-run rewards, just as in learning algorithms where regret guarantees vanish over time.

# E   OMITTED PROOFS FROM SECTION 3

## E.1   PROOF OF PROPOSITION 3.1

We prove Proposition 3.1.

*Proof of Proposition 3.1.* **Equilibrium profile.** Let $\mathcal{P} = \{0, \frac{1}{K}, \frac{2}{K}, \ldots, 1\}$ and define

$$q^* = \left\lfloor \frac{1}{N} \right\rfloor_K = \max\{x \in \mathcal{P} : x \leq \frac{1}{N}\}$$

Since the grid spacing is $1/K$, we have that

$$q^* \geq \frac{1}{N} - \frac{1}{K}$$

Now let $S^*$ be the profile where every player

$$\begin{cases} \text{posts price } 1 & \text{as long as nobody has ever priced below } 1, \\ \text{posts price } q^* & \text{if any player has priced below } 1. \end{cases}$$

While everyone plays 1, they tie and each get $\frac{1}{N}$ per round. From the round following the first deviation onwards, the deviator either prices above $q^*$ (payoff $0 < 1/N$), ties at $q^*$ (payoff $q^*/N \leq 1/N^2 < 1/N$), or undercuts to at most $q^* - \frac{1}{K} < 1/N$. Thus, if the first time a player prices below 1 is round $t \leq T$, the time-averaged utility after deviation is at most

$$\underbrace{\frac{(t-1)\frac{1}{N}}{T}}_{\text{before deviation}} + \underbrace{\frac{1}{T}}_{\text{round of deviation}} + \underbrace{\frac{(T-t)\frac{1}{N}}{T}}_{\text{after deviation}} \leq \frac{1}{N} + \frac{1}{T}$$

Therefore no deviation can improve on the payoff for profile $S^*$ by more than $\frac{1}{T}$, so $S^*$ is a $\frac{1}{T}$-approximate equilibrium.

**Defection by a no-regret learner.** Suppose player $i$ defects to a no-regret strategy $S_i^r$ with regret bound $r(T)$, while others continue playing $S_{-i}^*$. Then

$$\text{Price}(S_i^r, S_{-i}^*) \geq U_i(S_i^r, S_{-i}^*),$$

so it suffices to lower-bound player $i$'s utility.

First, we will consider the case where $2N > K$. In this case, $\frac{1}{N} - \frac{2}{K} - \frac{r(T)}{T} \leq \frac{2}{K} - \frac{2}{K} - \frac{r(T)}{T} \leq 0$. Since the price cannot go below 0, the condition is trivially satisfied.

For the remainder of this proof, consider the case where $K \geq 2N$. Note that this implies that $q^* \geq \frac{1}{2N} > 0$, and thus is a valid price to play.

Consider the transcript of play for the strategy profile $(S_i^r, S_{-i}^*)$. Players $j \neq i$ will always either simultaneously play 1 or simultaneously play $q^*$.[6] For player $i$, consider the fixed benchmark strategy "always price at $q^* - \frac{1}{K}$" played against this fixed transcript of opponent prices. On all rounds, pricing at $q^* - \frac{1}{K}$ undercuts them and wins, yielding payoff exactly $q^* - \frac{1}{K}$ each round. The no-regret guarantee then gives

$$\text{Price}(S_i^r, S_{-i}^*) \geq U_i(S_i^r, S_{-i}^*) \geq \left(q^* - \frac{1}{K}\right) \cdot T - r(T).$$

Since $q^* \geq \frac{1}{N} - \frac{1}{K}$, dividing by $T$ yields

$$\frac{\text{Price}(S_i^r, S_{-i}^*)}{T} \geq \frac{1}{N} - \frac{2}{K} - \frac{r(T)}{T},$$

as claimed. $\qquad\square$

### E.2 Proof of Theorem 3.5

To prove Theorem 3.5, we first show the following lemma, which upper bounds the number of rounds where a learner does not best respond in a pricing game in terms of the learner's regret. This lemma follows from the regret bound combined with an assumption on the gap.

**Lemma E.1** (Few non–best-response rounds). *Consider a two-player repeated pricing game with a learner and an opponent. The learner plays a no-regret strategy with external regret $r(T)$ and the opponent chooses i.i.d. prices $X_t \sim \mathcal{D}$ each round on the grid $\mathcal{P} = \{0, \frac{1}{K}, \ldots, 1\}$. For any grid price $p$ define the one-round* Bertrand *payoff*

$$u(p, \mathcal{D}) = p \Pr_{X \sim \mathcal{D}}[X > p] + \frac{p}{2} \Pr_{X \sim \mathcal{D}}[X = p].$$

*Assume there is a unique best price $p^\star \in \mathcal{P}$ and a gap $\delta > 0$ such that $u(p^\star, \mathcal{D}) \geq u(p, \mathcal{D}) + \delta$ for every $p \neq p^\star$. Let the learner possibly play mixed strategies $P_t$ over $\mathcal{P}$ in each round $t$. Define*

$$\mathcal{B} := \sum_{t=1}^{T} \left(1 - P_t(p^\star)\right)$$

---

[6]As the no-regret strategy is randomized, we cannot (and do not need to) know at what round the no-regret player first defects.

*to be the total probability mass assigned to non–best-response prices across the $T$ rounds. Then*

$$\mathcal{B} \leq \frac{r(T)}{\delta}.$$

*Proof.* Let $u(P_t, \mathcal{D}) := \sum_{p \in \mathcal{P}} P_t(p) \, u(p, \mathcal{D})$ denote the expected utility under the mixed action $P_t$. External regret implies

$$\sum_{t=1}^{T} u(P_t, \mathcal{D}) \geq \sum_{t=1}^{T} u(p^\star, \mathcal{D}) - r(T).$$

For any round $t$, by the $\delta$-gap assumption,

$$u(p^\star, \mathcal{D}) - u(P_t, \mathcal{D}) = \sum_{p \in \mathcal{P}} P_t(p) \left( u(p^\star, \mathcal{D}) - u(p, \mathcal{D}) \right) \geq \delta \left( 1 - P_t(p^\star) \right).$$

Summing over $t$ and rearranging yields $\delta \, \mathcal{B} \leq r(T)$, hence $\mathcal{B} \leq r(T)/\delta$. $\qquad\square$

Now, we are ready to prove Theorem 3.5.

*Proof of Theorem 3.5.* **Tweaked punishment distribution.** Let $P_{c,K}$ be the discretized ERD as defined in Definition C.1.

Create

$$P_{c,K,\varepsilon}^{high}\{1\} = (1-\varepsilon)P_{c,K}\{1\} + \varepsilon, \qquad P_{c,K,\varepsilon}^{high}\{x\} = (1-\varepsilon)P_{c,K}\{x\} \ (\textit{if } x < 1).$$

As in Lemma E.1, let

$$u(p) = p \Pr_{X \sim \mathcal{D}}[X > p] + \frac{p}{2} \Pr_{X \sim \mathcal{D}}[X = p].$$

The extra $\varepsilon$ on price 1 ensures that

$$p^\star := 1 - \tfrac{1}{K}$$

is the unique best price for $u$ in $\mathcal{P} = \{0, 1/K, \ldots, 1\}$ with respect to this distribution. After this adjustment, we can lower bound the gap between playing $p^*$ against $P_{c,K,\varepsilon}^{high}$ and playing any other price $p \neq p^*$ against it:

$$u(p^\star, P_{c,K,\varepsilon}^{high}) - u(p, P_{c,K,\varepsilon}^{high}) \geq \delta := p^* \left( (1-\varepsilon)(c - \tfrac{1}{K}) + \varepsilon \right) - p \left( (1-\varepsilon)(c + \tfrac{1}{K}) \right)$$

$$\geq \frac{\varepsilon - (1-\varepsilon)\tfrac{1}{K}}{2}$$

for every $p \neq p^\star$.

Now, let $c := \lfloor \tfrac{1}{N} - 2\delta \rfloor_K$.

**Equilibrium construction.** Match each player $i$ to a distinct player $\pi(i)$ $\pi(i) \neq i$. Every player $i$ posts price 1 unless $\pi(i)$ is the *first* deviator, in which case $i$ switches forever to draws from $P_{c,K,\varepsilon}^{high}$. On the cooperative path every round ties at 1, giving each player $1/N$. A single deviation triggers exactly one punisher's $P_{c,K,\varepsilon}^{high}$, while every other player continues pricing at 1; thus, again we can evaluate the prices and utilities of the defecting player and the punishing player as if they are engaged in a 2-player game. The deviator's best reply then earns $u(p^\star, P_{c,K,\varepsilon}^{high}) \leq c + \delta \leq 1/N$, except for in the exact round that they deviate. Thus, deviation can only lead to an increase of utility of $\frac{1}{T}$.

**Defection by a no-regret learner.** Let player $j$ defect to an strategy with regret $r(T)$. After the first undercut, only $\pi(j)$ draws $X_t \sim P_{c,K,\varepsilon}^{high}$; the other $N-2$ players continue at 1. Because $X_t$ is independent of $j$'s current action, the distribution of $X_t$ is i.i.d. across rounds. Thus, the price outcome of this game is equivalent to that of a 2-player repeated Bertrand pricing game where one player plays a no-regret strategy and the other plays $X_t$.

Apply Lemma E.1 with the gap $\delta$: the number of rounds the no-regret defector chooses $p_t \neq p^\star$ is $|\mathcal{B}| \leq r(T)/\delta$.

**Market price calculation.** Suppose that all players price at 1 except one player who prices at $P_{c,K,\varepsilon}^{high}$ and one player who prices at $1 - \frac{1}{K}$; in this case, the expected market price is exactly the expected value of $P_{c,K,\varepsilon}^{high}$, but moving all mass at price 1 to price $1 - \frac{1}{K}$. As $P_{c,K}$ is a $K$-discretization of the ERD (Appendix C) and since $P_{c,K,\varepsilon}^{high}$ stochastically dominates $P_{c,K}$, the expected value of $P_{c,K,\varepsilon}^{high}$ is at least $c(\ln(\frac{1}{c}) + 1) - \frac{1}{K}$. Thus in a *good* round ($p_t = p^\star$), the market price is at least

$$
\mathbb{E}_{X \sim P_{c,K,\varepsilon}^{high}}[\min(p^\star, X)] = \Pr(p^* \geq X) \cdot \mathbb{E}[X | X \leq p^*] + \Pr(p^* < X) \cdot p^*
$$
$$
= \Pr(p^* \geq X) \cdot \mathbb{E}[X | X \leq p^*] + \Pr(p^* < X) \cdot \mathbb{E}[X | X > p^*] + \Pr(p^* < X) \cdot p^* - \Pr(p^* < X) \cdot \mathbb{E}[X | X > p^*]
$$
$$
= \mathbb{E}[X] + \Pr(p^* < X) (p^* - \mathbb{E}[X | X > p^*])
$$
$$
= \mathbb{E}[X] + \Pr(p^* < X) (p^* - 1)
$$
$$
= \mathbb{E}[X] - \Pr(p^* < X) \left( \frac{1}{K} \right)
$$
$$
\geq \mathbb{E}[X] - \frac{1}{K}
$$
$$
\geq c(\ln(\frac{1}{c}) + 1) - \frac{2}{K} =: w_{\text{good}}
$$

In a *bad* round welfare is at least 0. Hence

$$
\text{DefectedPrice}(S_i^r, S_{-i}^*) \geq w_{\text{good}}(1 - \frac{|\mathcal{B}|}{T})
$$

Substituting $|\mathcal{B}| \leq r(T)/\delta$ and $c \geq \frac{1}{N} - 2\delta - \frac{1}{K}$ gives

$$
\text{DefectedPrice}(S_i^r, S_{-i}^*) \geq (1 - \frac{r(T)}{\delta T})((\frac{1}{N} - 2\delta - \frac{1}{K})(\ln(N) + 1) - \frac{2}{K})
$$
$$
\geq (\frac{1}{N} - 2\delta - \frac{1}{K})(\ln(N) + 1) - \frac{2}{K} - O\left( \frac{r(T)}{\delta T} \right)
$$

Recalling that $\delta = \Theta(\varepsilon)$, we may write the last term as $O(r(T)/T\epsilon)$. Thus, we get that

$$
\frac{\text{Price}(S_i^r, S_{-i}^*)}{T} \geq \frac{\ln(N) + 1}{N} - \frac{3\ln(N)}{K} - \Theta(\varepsilon) \cdot (\ln(N) + 1) - O\left( \frac{r(T)}{T\epsilon} \right).
$$

By setting $\varepsilon = \sqrt{\frac{r(T)}{T(\ln(N) + 1)}}$, we get

$$
\frac{\text{Price}(S_i^r, S_{-i}^*)}{T} \geq \frac{\ln(N) + 1}{N} - \frac{3\ln(N)}{K} - O\left( \sqrt{\frac{r(T)(\ln(N) + 1)}{T}} \right).
$$

$\square$

### E.3 PROOF OF PROPOSITION 3.2

*Proof of Proposition 3.2.* Let $\mathcal{S}^*$ be the equilibrium strategy profile where every player other than $i$ uses the following strategy:

$$
\begin{cases}
\text{posts price 1} & \text{as long as no player other than } i \text{ has ever priced below 1,} \\
\text{posts price 0} & \text{if any player other than } i \text{ has priced below 1.}
\end{cases}
$$

Player $i$ always posts price $1 - \frac{1}{K}$.

This is an $O(1/T)$-approximate equilibrium as long as $N \geq 3$: every player other than $i$ attains utility 0 regardless of whether they remain in the prescribed strategy or deviate, and player $i$ attains the highest feasible price of $1 - \frac{1}{K}$, so there is no unilateral deviation that increases the profit by $\omega(1/T)$.

Moreover, the strategies of all players other than $i$ are not responsive to the prices set by player $i$; therefore, when reasoning about the price outcomes under a deviation by player $i$, we can assume all others continue to price at 1.

If player $i$ deviates to a no-regret strategy $S_i^r$ with regret bound $r(T)$, the optimal fixed price in hindsight against opponents who always price at 1 remains at $1 - \frac{1}{K}$. By the no-regret guarantee, the time-averaged utility of player $i$ satisfies

$$U_{i*}(S_i^r, S_{-i}^*) \; \geq \; 1 - \frac{1}{K} - \frac{r(T)}{T} - O\left(\frac{1}{T}\right),$$

where the $O(1/T)$ term accounts for at most a single transient round.

Since the utility of a single player lower bounds the market price in each round, the time-averaged market price is at least

$$1 - \frac{1}{K} - \frac{r(T)}{T} - O\left(\frac{1}{T}\right).$$

$\square$

### E.4 Proof of Lemma 3.4

The proof of Lemma 3.4 uses the no-regret property of the defection strategy $S_i$ to lower bound the utility of player $i$ by the static best-response to the time-averaged distribution of other agents' prices. The remainder of the argument boils down to bounding the market price in a static game ($T = 1$) in terms of the utility achieved by a best-responding agent. To prove this, we show that in a pricing game, a no-regret strategy is guaranteed a fraction of the total welfare that is independent of the number of players $N$.

We formalize this as follows.

*Proof of Lemma 3.4.* Consider any no-regret strategy $S_i^r$ with regret bound $r(T)$ and any other $N - 1$ strategies $S_{-i}$. It suffices to prove the statement for $c := U_i(S_i^r, S_{-i})/T$, the time-averaged utility of player $i$ given this strategy profile. Let $\mathcal{D}$ denote the distribution over $T$-length sequences of $N$-player prices given $(S_i^r, S_{-i})$. Finally, let $\mathcal{D}_{-i}$ denote the distribution over $T$-length sequences of min prices $\min_{j \neq i} P_{i,t}$ of all players except for player $i$ on this same strategy profile, and let $\mathcal{D}_{-i}^t$ denote the distribution of prices only on day $t$.

We split the proof into two steps. First, we show that $\frac{1}{T}\mathcal{D}_i^t$ is approximately stochastically dominated by an equal-revenue distribution with parameter $c$. Then, we use this to lower bound the market price.

**Step 1: Approximate Stochastic Dominance.** By the regret guarantee of $S_i$, we have

$$c \cdot T = U_i(S_i^r, S_{-i}) \; \geq \; \max_p \sum_{t=1}^{T} \mathbb{E}[\, u_i(p, P_{-i,t})\,] - r(T),$$

where $u_i$ is the Bertrand payoff as defined in Section 2. Since $u_i(p, P_{-i,t}) \geq p \cdot \Pr[p < \min_{j \neq i} P_{j,t}] = \Pr[p < \mathcal{D}_{-i}^t]$, it follows that

$$c \cdot T \; \geq \; \max_p \sum_{t=1}^{T} p \cdot \Pr\big(p < \mathcal{D}_{-i}^t\big) - r(T).$$

Thus, for all $p \in \{0, \frac{1}{K}, \ldots, 1\}$,

$$\frac{1}{T} \sum_{t=1}^{T} \Pr\big(\mathcal{D}_{-i}^t > p\big) \; \leq \; \frac{c}{p} + \frac{r(T)}{T \cdot p}.$$

This shows that the empirical mixture distribution $\frac{1}{T}\mathcal{D}^t_{-i}$ is approximately stochastically dominated by an equal-revenue distribution with parameter $c$ (Appendix C).

**Step 2: Upper Bound Market Price.** Let $c'$ be the smallest number $\geq c$ such that $Kc'$ is an integer. Thus, $c \leq c' \leq c + \frac{1}{K}$. We can now upper bound the cumulative market price by

$$
\begin{aligned}
& T \cdot \mathrm{Price}(S_i^r, S_{-i}) \\
&= \mathbb{E}\Big[\sum_{t=1}^{T} \min_{j}(P_{j,t})\Big] \\
&\leq \mathbb{E}\Big[\sum_{t=1}^{T} \min_{j \neq i}(P_{j,t})\Big] \\
&= \sum_{t=1}^{T} \mathbb{E}[\mathcal{D}^t_{-i}] \\
&\leq \sum_{t=1}^{T}\sum_{j=1}^{K} \mathrm{Pr}\Big(\mathcal{D}^t_{-i} > \tfrac{j}{K}\Big)\frac{1}{K} \\
&= \sum_{j=K\cdot c'}^{K}\sum_{t=1}^{T} \mathrm{Pr}\Big(\mathcal{D}^t_{-i} > \tfrac{j}{K}\Big)\frac{1}{K} + \sum_{t=1}^{T}\sum_{j=1}^{K\cdot c'-1} \mathrm{Pr}\Big(\mathcal{D}^t_{-i} > \tfrac{j}{K}\Big)\frac{1}{K} \\
&\leq \sum_{j=K\cdot c'}^{K} \frac{K\big(c\cdot T + r(T)\big)}{j}\frac{1}{K} + \sum_{t=1}^{T}\sum_{j=1}^{K\cdot c'-1} \mathrm{Pr}\Big(\mathcal{D}^t_{-i} > \tfrac{j}{K}\Big)\frac{1}{K} && \text{(By the result above)} \\
&\leq \sum_{j=K\cdot c'}^{K} \frac{c\cdot T + r(T)}{j} + \sum_{t=1}^{T}\sum_{j=1}^{K\cdot c'-1} \frac{1}{K} \\
&\leq (c\cdot T + r(T))\Bigg(\sum_{j=K\cdot c'}^{K}\frac{1}{j}\Bigg) + c'\cdot T \\
&\leq (c\cdot T + r(T))\Big(1 + \ln\Big(\tfrac{1}{c'}\Big)\Big) + c'\cdot T && \text{(By the Harmonic-series bound)} \\
&\leq (c\cdot T + r(T))\Big(1 + \ln\Big(\tfrac{1}{c}\Big)\Big) + \Big(c + \tfrac{1}{K}\Big)\cdot T
\end{aligned}
$$

Thus, if the no-regret player attains utility $c \cdot T$ over all rounds, the total expected minimum price is at most $(c \cdot T + r(T))\left(1 + \ln(\frac{1}{c})\right) + \left(c + \frac{1}{K}\right) \cdot T$. Dividing all terms by $T$ gives us our result.

$\square$

### E.5    PROOF OF THEOREM 3.3

We prove Theorem 3.3 using Lemma 3.4.

*Proof of Theorem 3.3.* First, note that as the total welfare per round is upper bounded by 1, any player $i$ in the median profit set must have total welfare in $S^*$ upper bounded by $\frac{2T}{N}$. Furthermore, as $S^*$ is a $\epsilon$-approximate equilibrium, it must be that if this player $i$ instead played some $r(T)$-regret strategy $S_i^r$ against $S^*_{-i}$, they would achieve utility at most $\frac{2T}{N} + \epsilon \cdot T$—otherwise this would violate our equilibrium assumption. Thus, $U_i(S_i^r, S^*_{-i}) \leq \frac{2T}{N} + \epsilon \cdot T$. Applying Lemma 3.4 with $c = 2/N + \epsilon$,

the total average minimum price is upper bounded by

$$\text{Price}(S_i^r, S_{-i}^*) \leq \frac{2}{N} + \epsilon + \frac{1}{K} + \left(\frac{2}{N} + \epsilon + \frac{r(T)}{T}\right)\left(\ln(\frac{1}{\frac{2}{N} + \epsilon}) + 1\right)$$

$$< \frac{2}{N} + \epsilon + \frac{1}{K} + \left(\frac{2}{N} + \epsilon + \frac{r(T)}{T}\right)(\ln(N) + 1)$$

$$= \epsilon + \frac{1}{K} + \frac{2}{N}(\ln(N) + 2) + \left(\epsilon + \frac{r(T)}{T}\right)(\ln(N) + 1)$$

$$= \frac{2\ln(N) + 4}{N} + \left(O(\frac{1}{T}) + \frac{r(T)}{T}\right)(\ln(N) + 1) + O(\frac{1}{T}) + \frac{1}{K}$$

$$= O\left(\frac{2\ln(N) + 4}{N} + \frac{r(T) + 1}{T}\ln(N) + \frac{1}{T} + \frac{1}{K}\right)$$

We have shown the result for at least one player $i$, completing the proof. □

### E.6   PROOF OF THEOREM 3.6

The proof of Theorem 3.6 again uses Lemma 3.4.

*Proof of Theorem 3.6.*  Our goal is to upper bound $\text{DefectedPrice}(I, S, \{S_i^r\}_{i \in I})$. For brevity, let us denote this value as $D$ for the remainder of this proof. Let $\hat{S}$ be the strategy profile corresponding to the players in $I$ defecting to $\{S_i^r\}_{i \in I}$. As there are at least $M$ no-regret players in this profile, it must be that there is some no-regret player $i$ who has total utility upper-bounded by $\frac{D}{M}$. Thus, $U_i(\hat{S}) \leq \frac{D}{M}$. We will consider two cases:

- $D \leq \frac{M}{e^{M-1}}$. In this case, we are done.

- $D \geq \frac{M}{e^{M-1}}$. We can apply Lemma 3.4 with $c = D/M$ to upper bound the average price after defection as follows:

$$D \leq \frac{D}{M} + \frac{1}{K} + \left(\frac{D}{M} + \frac{r(T)}{T}\right)\ln\left(\frac{M}{D}\right)$$

$$\implies \frac{\frac{M-1}{M}D - \frac{1}{K}}{\frac{D}{M} + \frac{r(T)}{T}} \leq \ln\left(\frac{M}{D}\right)$$

$$\implies D \leq me^{-\frac{\frac{M-1}{M}D - \frac{1}{K}}{\frac{D}{M} + \frac{r(T)}{T}}}$$

$$\implies D \leq me^{-\frac{\frac{(M-1)}{M} - \frac{1}{KD}}{\frac{1}{M} + \frac{r(T)}{TD}}}$$

$$\implies D \leq me^{-\frac{M-1-\frac{e^{M-1}}{K}}{1 + \frac{r(T)e^{M-1}}{T}}},$$

where the last step uses the fact that $D \geq M/e^{M-1}$.

Let $\varepsilon_1 = \frac{e^{M-1}}{K}$, and let $\varepsilon_2 = \frac{r(T)e^{M-1}}{T}$. Note that

$$\frac{M - 1 - \varepsilon_1}{1 + \varepsilon_2} \geq (M - 1 - \varepsilon_1)\left(1 - \varepsilon_2\right) \qquad \text{(By the fact that } \frac{1}{1+\varepsilon_2} \geq 1 - \varepsilon_2\text{)}$$

$$= (M - 1) - \varepsilon_2(M - 1) - \varepsilon_1 + \varepsilon_1\varepsilon_2$$

$$\geq (M - 1) - \varepsilon_2(M - 1) - \varepsilon_1$$

Therefore,

$$me^{-\frac{M-1-\frac{e^{M-1}}{K}}{1+\frac{r(T)e^{M-1}}{T}}} \leq me^{-(M-1)+\frac{r(T)e^{M-1}}{T}(M-1)+\frac{e^{M-1}}{K}}$$

$$= \frac{M}{e^{M-1}} \cdot e^{\frac{r(T)e^{M-1}}{T}(M-1)+\frac{e^{M-1}}{K}}$$

$$= \frac{M}{e^{M-1}} \cdot (1 + o_T(1) + o_K(1))$$

$\square$

### E.7 STATEMENT AND PROOF OF PROPOSITION E.2

We state and prove Proposition E.2.

**Proposition E.2.** *Let $N \geq M \geq 2$, $K > 1$, and $T \geq 1$. There exists an $O(1/T)$–approximate repeated game equilibrium strategy profile $S^*$ with market price of 1 and a collection of $M$ no-regret strategies $S_1^r, \ldots, S_M^r$ with regret $o_{\min(T,K)}(1)$ such that for any subset $I \subseteq [N]$ of size $M$, if the players in $I$ defect by adopting the strategies $S_1^r, \ldots, S_M^r$, then it holds that:*

$$DefectedPrice(I, S^*, \{S_j^r : j = 1, \ldots, M\}) \geq \frac{M}{e^{M-1}} - O\left(\frac{1}{K}\right).$$

Our analysis shows that the same bound from (Feldman et al., 2016) applies even in the presence of other agents that choose arbitrary strategies. The proof of Proposition E.2 follows from the coarse-correlated equilibrium construction coupled with constructing the other agents so that they choose a price of 1 at every round. Our regret bound is on the order of $o_{\min(T,K)}(1)$ rather than $o_T(1)$ because we must account for theregret incurred by discretizing continuous distributions to the $\frac{1}{K}$ grid.

*Proof.* **Equilibrium profile $S^*$.** All players follow the same repeated-game strategy $S^*$:

- By default, play price 1 at every round.

- If at some round multiple players simultaneously play below 1 for the first time, then keep playing 1 forever.

- If at some round there is just a *single* player who prices below 1, and in all previous rounds all players priced at 1, then from the next round onward all players switch to posting price 0 forever (punishment phase).

$S^*$ **is an approximate equilibrium.** Fix a player $i$ and suppose they deviate unilaterally from $S^*$. Let $t$ be the (first) round in which $i$ posts a price $< 1$ while all other players still post 1. If no such $t$ exists, then $i$ receives $\frac{1}{N}$ per round exactly as under $S^*$. If such a $t$ exists with $t \leq T$, then in round $t$ player $i$ can earn at most 1 (by undercutting), and from round $t+1$ onward the punishment phase is triggered and all players play 0, so $i$ receives zero thereafter. Hence $i$'s total payoff is at most

$$(t-1) \cdot \frac{1}{N} + 1 \leq T \cdot \frac{1}{N} + 1,$$

and their time-averaged payoff is at most $\frac{1}{N} + \frac{1}{T}$. By following $S^*$ they receive exactly $\frac{1}{N}$ per round, so any unilateral deviation improves utility by at most $1/T$. Thus $S^*$ is a $1/T$–approximate equilibrium.

**Coarse-correlated equilibrium construction.** By Feldman et al. (2016), there exists a distribution $\mathcal{H}$ on $(0, 1]$ such that:

(i) If $M$ players play i.i.d. from $\mathcal{H}$, then every fixed price obtains the same expected utility in the support of $\mathcal{H}$ (so $\mathcal{H}$ is a coarse correlated equilibrium).

(ii) The market price is exactly $\frac{M}{e^{M-1}}$.

We now discretize this equilibrium construction. Let $\mathcal{H}^{(K)}$ denote the distribution obtained by drawing $x \sim \mathcal{H}$ and replacing it with $\lfloor Kx \rfloor / K$, with the convention that 1 is mapped to $1 - 1/K$. Observe that the expected utilities and the benchmark utilities of all players differ from $\mathcal{H}$ by at most $O(1/K)$. This implies that:

- $\mathcal{H}^{(K)}$ is an $(2/K)$-approximate coarse-correlated equilibrium

- the expected market price in a one-round game where all $M$ players play $\mathcal{H}^{(K)}$ is at least $\frac{M}{e^{M-1}} - O(1/K)$.

Note that $\mathcal{H}^K$ has no support on price 1.

**Construction of no-regret defectors.** The no-regret strategies will operate as follows: they will play $\mathcal{H}^{(K)}$ each round as long as their regret remains below $\frac{2}{K}$. Otherwise, they will defect to a standard no-regret strategy such as multiplicative weights.

**Plugging defectors into $S^*$.** Suppose a set $I$ of $M \geq 2$ players deviates to the i.i.d. strategies $\hat{S}_1^r = \cdots = \hat{S}_M^r$. These strategies have no support at any round on the price 1. So on the very first deviation round, all defectors post prices $< 1$. Thus this is classified as a multi-deviation and the punishment phase is not triggered. All non-deviators continue posting 1 forever. The market price is determined solely by the $M$ defectors, whose expected price is at least $\frac{M}{e^{M-1}} - O(1/K)$.

This completes the proof. $\qquad\square$

# F    ADDITIONAL CONTENT FROM SECTION 4.2

We formally define a *defection-aware* equilibrium below.

**Definition F.1.** *Let $N \geq 2$, and let $i \in [N]$. Let $S_i$ be any strategy. A strategy profile $(S_1^*, \ldots, S_{i-1}^*, S_{i+1}^*, \ldots, S_N^*)$ is a **defection aware repeated-game equilibrium with respect to** $(i, S_i)$ if it is a Nash equilibrium in strategy space in the $N - 1$ player game between players in $J = [N] \setminus \{i\}$, with utility functions given by:*

$$\tilde{U}_j(S_j, S_{J \setminus \{j\}}) := U_j(S_j, S_{[N] \setminus \{j\}}).$$

*We also consider $\epsilon$-relaxations of this equilibrium concept.*

## F.1    PROOF OF THEOREM 4.1

We prove Theorem 4.1.

*Proof of Theorem 4.1.* **Equilibrium strategies.** Let $J = [N] \setminus \{i\}$. For each $j \in J$, define the strategy $S_j^*$ as follows:

- At round $t$, if in all previous rounds every player in $J$ (including $j$) has posted price 1, then play 1.

- Otherwise (i.e., if some player in $J$ has ever posted a price $< 1$), then from the next round onward play 0 forever.

The designated player $i$ is ignored in this condition: $i$'s actions never trigger punishment. Let $\mathcal{S}_{-i}$ denote the strategy profile of the players in $J$.

**$\mathcal{S}_{-i}$ is an approximate defection-aware equilibrium with respect to $i$.** Fix $j \in J$ and consider a unilateral deviation by $j$, holding the other players in $J$ to $\mathcal{S}_{-i}$ (player $i$ plays $S_i^r$ throughout). Let $t$ be the first round in which $j$ posts a price $< 1$ while all other players in $J$ still post 1. If no such $t$ occurs, $j$'s average payoff equals their payoff in the equilibrium. If such $t \leq T$ exists, then in round $t$ the most $j$ can earn is 1 (by undercutting). From round $t + 1$ onward, the punishment rule is triggered and all players play 0, so $j$ earns 0 thereafter. Hence $j$'s total payoff is at most their payoff in the equilibrium plus 1. Thus any unilateral deviation by any $j$ gains at most $1/T$ on average, so $\mathcal{S}_{-i}$ is a $1/T$–approximate defection-aware equilibrium with respect to $i$.

**Market price outcome.** Along the equilibrium path, all players in $J$ post 1 in every round (since no player in $J$ deviates). By construction, the players in $J$ ignore agent $i$'s play in the trigger condition. Thus the per-round market price is simply $\min\{1, p_{h,t}\} = p_{h,t}$. That is, the market price is fully determined by player $i$'s strategy $S_i^r$.

Since $S_i^r$ is no-regret, its average payoff is within $r(T)/T$ of the best fixed action against opponents at constant price 1. The best fixed price in this environment is $1 - \frac{1}{K}$ (the largest grid point below 1), which wins every round. Therefore the realized average market price satisfies

$$\mathrm{Price}\big(\mathcal{S}_{-i},\, S_i^r\big) \;\; \geq \;\; 1 - \frac{1}{K} \;-\; \frac{r(T)}{T}.$$

**Conclusion.** $\mathcal{S}_{-i}$ is an $O(1/T)$–approximate defection-aware equilibrium with respect to $(i, S_i^r)$, and in the presence of the no-regret player $i$, the market price remains bounded below as claimed. $\quad\square$

### F.2 ALTERNATE CONSTRUCTION OF DEFECTION-AWARE EQUILIBRIA WITH WELFARE GUARANTEES

Since the market price is equal to the social welfare, it is also natural to ask how the welfare is distributed between AI agents and humans. In the equilibrium construction of Theorem 4.1, it turns out that the AI agents cede all of the welfare gain to the human. To address this degeneracy, we show in Appendix F.2 that there are also defection-aware equilibria that both induce high prices and give the AI agents at least a $\Omega_{K,N}(1)$ share of those prices.

**Theorem F.2.** *Let $N \geq 4$, $K > 1$, and $T \geq 1$. Fix a designated player $i \in [N]$ who runs a no-regret strategy $S_i^r$ with regret $r(T)$. Then there exists an $O(1/T)$–approximate defection-aware repeated-game equilibrium $\mathcal{S}_{-i}$ with respect to $(i, S_i^r)$ such that:*

$$Price(\mathcal{S}_{-i}, S_i^r) \geq \frac{1}{T}\sum_{t=1}^{T} \sum_{j \in [N]\setminus\{i\}} \mathbb{E}\big[\, u_j(p_{j,t}, p_{-j,t})\,\big] \;\; \geq \;\; \Omega_{K,N}(1)$$

Thus, even as $K$ and $N$ increase, the AI agent welfare remains a constant.

*Proof.* **Equilibrium construction.** Let $J = [N] \setminus \{i\}$ and fix a leader $\ell \in J$. Each follower $j \in J \setminus \{\ell\}$ plays price 1 every round unless some other player in $J \setminus \{\ell\}$ deviates from its prescribed strategy; if any player in $J \setminus \{\ell\}$, then from the next round onward all players in other player in $J \setminus \{\ell\}$ play 0 forever (punishment). The human's actions never trigger punishment.

Note first that this is an $(1/T)$-approximate equilibrium, regardless of what the $\ell$ does; the followers are mutually threatening each other to continue to price at 1. Thus, the problem reduces to a 2-player game in which the human is playing no-regret, and $\ell$ can choose any strategy to maximize their utility.

By Theorem 4.7 in Arunachaleswaran et al. (2025), the utility of $\ell$ in this setting can be $\Omega_K(1)$, via an ERD-based strategy. We can lower bound the cumulative utility of all of the players in $J$ by the utility of this player. $\quad\square$

## G   ADDITIONAL CONTENT FOR SECTION 4.3

### G.1   PROOF OF THEOREM 4.2

**Tie-breaking convention** Throughout this section, when a defector uses Follow-The-Leader (FTL), any ties at the chosen price are resolved in favor of the FTL player. Under this convention, the discretized equal-revenue distribution of Definition C.1 equalizes payoffs *exactly* at all grid prices.

*Proof of Theorem 4.2.* In the equilibrium profile, each agent $i$ is matched to a distinct agent to punish $\pi(i) \neq i$. Agent $i$ employs the following strategy:

$$\begin{cases} \text{posts price 1} & \text{as long as } \pi(i) \text{ has never priced below 1, or was not the first to price below 1,} \\ \text{plays the sequence } \phi & \text{if } \pi(i) \text{ was the first to price below 1, and } \pi(i)\text{'s actions are consistent with an FTL strategy,} \\ \text{posts price 0} & \text{otherwise} \end{cases}$$

This ensures that, if exactly one agent defects, all agents except for one continue pricing at 1, and exactly one agent punishes the defector. If the defector behaves according to FTL, they will be punished according to $\phi$. If they ever deviate from FTL behavior, they will be punished by the punisher playing price 0 forever.

In order to define the sequence $\phi$ we must first define the following distribution $\mathcal{P}^{\text{low}}_{\bar{c},K,\epsilon}$, for $\varepsilon < \frac{1}{KT}$:

Let $c = \sqrt{1/N - 1/K}$. Furthermore, let $\bar{c} := \lfloor Kc \rfloor / K$ (so $\bar{c} \in \mathcal{P}$ and $\bar{c} \leq c < \bar{c} + 1/K$). On $\mathcal{P} = \{0, \frac{1}{K}, \ldots, 1\}$, we first define the baseline discretized equal revenue distribution (ERD) for $\bar{c}$. This distribution, $P_{\bar{c},K}$, is formally defined in Definition C.1. The distribution $\mathcal{P}^{\text{low}}_{\bar{c},K,\epsilon}$ is supported on $\{\bar{c}, \bar{c} + 1/K, \bar{c} + 2/K, \ldots, 1\}$ (Def. C.1) and shifting an $\varepsilon$ mass to $\bar{c} + 1/K$ so that $p^\star = \bar{c}$ is the unique maximizer of $u(\cdot)$. Write $m(\bar{c})\{x\}$ for the ERD probability mass at price $x$.

Formally:

$$\mathcal{P}^{\text{low}}_{\bar{c},K,\epsilon}\{\bar{c} + \frac{1}{K}\} = (1-\varepsilon)m(\bar{c})\{\bar{c} + \frac{1}{K}\} + \varepsilon\bar{c}, \qquad \mathcal{P}^{\text{low}}_{\bar{c},K,\epsilon}\{x\} = (1-\varepsilon)m(\bar{c})\{x\} \; (x > \bar{c} + \frac{1}{K}).$$

Recall that we are modeling the FTL player as having ties broken in their favoe. Thus the extra $\varepsilon$ on price $\bar{c} + 1/K$ ensures that the price

$$p^\star := \bar{c}$$

is the unique best price in the grid in response for $u$ with respect to the distribution $\mathcal{P}^{\text{low}}_{\bar{c},K,\epsilon}$, and in fact gets utility of $\bar{c}$. A direct computation shows that

$$u(p \neq p^*, \mathcal{P}^{\text{low}}_{\bar{c},K,\epsilon}) = (1-\varepsilon)\bar{c}$$

The sequence $\phi$ will be constructed as follows:

**Construction of the sequence $\phi$.** The sequence $\phi$ is defined to playing a cycle $\phi_1, \phi_2, \ldots, \phi_L, \phi_1, \ldots$ until the end of the game, where each $\phi_j$ is a price distribution. We take the cycle length to be

$$L := \left\lceil \frac{1}{\bar{c}} \right\rceil.$$

We define the components of this cycle as follows. Write the CDF of $\mathcal{P}^{\text{low}}_{\bar{c},K,\epsilon}$ as

$$F(x) = \Pr_{X \sim \mathcal{P}^{\text{low}}_{\bar{c},K,\epsilon}}[X \leq x].$$

For $j = 0, 1, \ldots, L$ define the $j$th quantile threshold

$$q_0 := \bar{c}, \qquad q_j := \inf\{x \in \mathcal{P} : F(x) \geq j/L\}, \; j = 1, \ldots, L.$$

Now slice $\mathcal{P}^{\text{low}}_{\bar{c},K,\epsilon}$ into the $L$ equal-mass "chunk" distributions

$$\phi_j(x) = \begin{cases} \dfrac{\mathcal{P}^{\text{low}}_{\bar{c},K,\epsilon}(x)}{F(q_j) - F(q_{j-1})}, & x \in \{q_{j-1}, q_{j-1} + \frac{1}{k}, \ldots, q_j\}, \\ 0, & \text{otherwise}, \end{cases} \quad j = 1, \ldots, L.$$

Each $\phi_j$ is supported on $\{q_{j-1}, q_{j-1} + 1/k, \ldots, q_j\}$, so its sub-minimum support point is

$$m_j := \min\{x : \phi_j(x) > 0\} = q_{j-1},$$

and in particular

$$m_1 \geq m_2 \geq \cdots \geq m_L = q_0 = \bar{c},$$

**FTL Dynamics against $\phi$** We now establish that for every cycle number $\tau \geq 1$ and within-cycle step $s \in \{1, \ldots, L\}$, letting

$$t = (\tau - 1)L + s,$$

the FTL learner's choice satisfies

$$\begin{cases} p_t = \bar{c}, & s = 1, \\ p_t \geq m_s, & 2 \leq s \leq L. \end{cases}$$

Fix $\tau, s$ and set $t = (\tau-1)L+s$. Compare three candidates: $\bar{c}$, $m_s$, and any $p$ such that $\bar{c} < p < m_s$. Let $R_t(p)$ denote the total reward of price $p$ against the first $t$ steps of the sequence $\phi$. Note that at time $t$, FTL operates by playing $\arg\max_{p \in \mathcal{P}} R_{t-1}(p)$.

Case 1: $s = 1$. Then $t - 1 = (\tau-1)L$ and no "within-cycle" terms appear:

$$R_{t-1}(p) = \sum_{r=1}^{(\tau-1)L} p\,\mathbf{1}\{p < X_r\} = (\tau-1)L\,u(p, \mathcal{P}^{\text{low}}_{\bar{c},K,\epsilon}).$$

Since $u(\bar{c}, \mathcal{P}^{\text{low}}_{\bar{c},K,\epsilon}) = \bar{c} > (1-\varepsilon)\bar{c} = u(p, \mathcal{P}^{\text{low}}_{\bar{c},K,\epsilon})$ for all $p \neq \bar{c}$, the unique maximizer is $p_t = \bar{c}$.

Case 2: $2 \leq s \leq L$. Write

$$R_{t-1}(p) = (\tau-1)L\,u(p, \mathcal{P}^{\text{low}}_{\bar{c},K,\epsilon}) + \sum_{r=(\tau-1)L+1}^{(\tau-1)L+s-1} p\,\mathbf{1}\{p < X_r\}.$$

In rounds $(\tau-1)L+1, \ldots, (\tau-1)L+s-1$, the learner has faced exactly the distributions $\phi_1, \ldots, \phi_{s-1}$. Note that as all prices in the support so far in the cycle are strictly above $m_s$, any $p < m_s$, each of those $s-1$ rounds contributes $p$, whereas for $p = m_s$ it contributes exactly $m_s$. Moreover $u(p, \mathcal{P}^{\text{low}}_{\bar{c},K,\epsilon}) = u(m_s, \mathcal{P}^{\text{low}}_{\bar{c},K,\epsilon}) = (1-\varepsilon)\bar{c}$ for all full-cycle terms. Thus, $R_{t-1}(m_s) > R_{t-1}(p)$, for $c < p < m_s$. Now, we need only compare $m_s$ and $\bar{c}$. The utility of $\bar{c}$ is exactly $\bar{c}$ times the number of rounds so far, as all prices in the cycle are strictly above $\bar{c}$. Thus,

$$R_{t-1}(\bar{c}) = (\tau-1)L\bar{c} + \sum_{r=(\tau-1)L+1}^{(\tau-1)L+s-1} \bar{c}.$$

So,

$$R_{t-1}(m_s) - R_{t-1}(\bar{c}) = (\tau-1)L(1-\varepsilon)\bar{c} + \left(\sum_{r=(\tau-1)L+1}^{(\tau-1)L+s-1} m_s\right) - (\tau-1)L\bar{c} + \sum_{r=(\tau-1)L+1}^{(\tau-1)L+s-1} \bar{c}$$

$$= -(\tau-1)L\varepsilon + (s-2)(m_s - \bar{c})$$

$$\geq \frac{1}{K} - T\varepsilon$$

$$> 0 \qquad \qquad \text{(By the fact that } \varepsilon < \frac{1}{KT})$$

But $(T-1)\varepsilon$ is exactly the maximum possible lead that $\bar{c}$ could have built up over $m_s$ in the first $(\tau-1)L$ rounds, so in particular

$$R_{t-1}(m_s) > R_{t-1}(\bar{c}) \quad \text{and} \quad R_{t-1}(m_s) > R_{t-1}(p) \quad \forall p < m_s.$$

Thus the maximizer $p_t$ must satisfy $p_t \geq m_s$. This further implies that in every cycle, the FTL player attains utility only in the very first round, when then attain utility $\bar{c}$. Thus, over a $T$-round game, FTL can attain utility at most $\bar{c}^2$.

**Equilibrium Analysis.** Here we show that no deviation yields utility above the cooperative benchmark. Suppose player $i$ plays some strategy which is not pricing at 1 in every round. Then they must deviate from pricing at 1 at some round $\tau_0 + 1$. Furthermore, they must stop mimicking FTL for the first time at some round $\tau_1$, or if they never do, let $\tau_1 = T + 1$. We can thus partition the payoff of player $i$ into five parts:

$$U_i^{\text{dev}} = U^{\text{coop}} + U^{\text{FTL-first}} + U^{\text{FTL-rest}} + U^{\text{switch}} + U^{\text{punished}}.$$

During the rounds in which player $i$ continues to follow FTL, the punisher plays the cycle $\varphi_1, \ldots, \varphi_L$ with $L = \Theta(\sqrt{N})$ and parameter $c \approx 1/\sqrt{N}$. In each cycle the FTL learner earns a positive payoff only in the first round, when it plays $p = \bar{c}$ and wins at price $\bar{c}$, while in all other $L-1$ rounds the punisher draws a price in $[m_s, q_s]$ and undercuts. Thus the total payoff per cycle is at most $\bar{c}$, and the average payoff per round within the cycle is at most $\bar{c}/L \approx (1/\sqrt{N})/\sqrt{N} = 1/N$. Hence across the entire block of $\tau_1 - \tau_0$ rounds, the deviator's total payoff is bounded by $(\tau_1 - \tau_0)/N$.

Each term is bounded as follows:

$$U^{\mathrm{coop}} \leq \frac{\tau_0}{N},$$

$$U^{\mathrm{FTL-first}} \leq 1,$$

$$U^{\mathrm{FTL-rest}} \leq \frac{\tau_1 - \tau_0}{N},$$

$$U^{\mathrm{switch}} \leq 1,$$

$$U^{\mathrm{punished}} = 0.$$

Summing gives

$$U_i^{\mathrm{dev}} \;\leq\; \frac{T}{N} + 2,$$

so the time-averaged payoff for any deviation is

$$\frac{U_i^{\mathrm{dev}}}{T} \;\leq\; \frac{1}{N} + O\!\left(\frac{1}{T}\right).$$

**Expectation of the cycle distribution.** We record the expectation of the adjusted discretized ERD used for the cycle. Recall $\bar{c} = \lfloor Kc \rfloor / K$, and $\mathcal{P}_{\bar{c},K,\epsilon}^{\mathrm{low}}$ is obtained from the discretized ERD at parameter $\bar{c}$ by moving an $\varepsilon$-mass to $\bar{c} + 1/K$ (with $\varepsilon < 1/(KT)$). Then

$$\mathbb{E}\big[\mathcal{P}_{\bar{c},K,\epsilon}^{\mathrm{low}}\big] \;=\; (1-\varepsilon)\,\mathbb{E}\big[\mathrm{ERD}_{\mathrm{disc}}(\bar{c})\big] \,\pm\, O\!\big(\tfrac{1}{K}\big) \;\geq\; \bar{c}(H_{K-1} - H_{\bar{c}K-1}) - O\!\big(\tfrac{1}{K}\big) - O(\varepsilon), \quad (1)$$

where $H_n$ is the $n$th harmonic number and the $O(1/K)$ term is the discretization error between the continuous and grid ERD expectations.[7] Using $H_n \geq \ln n$ and $\bar{c} \asymp 1/\sqrt{N}$ gives

$$\mathbb{E}\big[\mathcal{P}_{\bar{c},K,\epsilon}^{\mathrm{low}}\big] \;\geq\; \bar{c}\ln\frac{1}{\bar{c}} \;-\; O\!\big(\tfrac{1}{K}\big) \;-\; O(\varepsilon) \;=\; \Omega\!\big(\tfrac{\ln N}{\sqrt{N}}\big).$$

*Cycle cutoff.* If $T$ is not a multiple of $L = \lceil 1/\bar{c} \rceil$, the last incomplete cycle contributes at most $O(1/L) = O(\bar{c})$ slack to the per-cycle average (or $O(1/T)$ to the time average). Grid snapping may unbalance chunk masses by $O(1/K)$, which affects the per-cycle average by at most $O\!\big(\tfrac{1}{KL}\big) = O\!\big(\tfrac{\bar{c}}{K}\big)$; both are absorbed by the error terms above.

**Market Price Calculation.** Now we analyze the market price when player $j$ defects to FTL and is punished by $\pi(j)$ playing the cycle $\phi$. As established above, the FTL player's price $p_t$ is at least $m_s$ for steps $s = 2, \ldots, L$ of each cycle. The punisher plays prices drawn from $\phi_s$, which are supported on $[q_{s-1}, q_s] \cap \mathcal{P}$. The market price in these rounds is $\min(p_t, X_t)$ where $X_t \sim \phi_s$. Since $p_t \geq m_s = q_{s-1}$, the market price in steps $2, \ldots, L$ is simply the price drawn by the punisher, $X_t$. In step $s=1$ of each cycle, $p_t = \bar{c}$ and $X_t \sim \phi_1$ has support strictly above $\bar{c}$, so the market price is $\min(\bar{c}, X_t) = \bar{c}$. Thus

$$\frac{1}{L}\Big(\bar{c} + \sum_{s=2}^{L} \mathbb{E}[\phi_s]\Big) = \mathbb{E}[\mathcal{P}_{\bar{c},K,\epsilon}^{\mathrm{low}}] - \frac{\mathbb{E}[\phi_1] - \bar{c}}{L} \;\geq\; \mathbb{E}[\mathcal{P}_{\bar{c},K,\epsilon}^{\mathrm{low}}] - \frac{1}{L} \;\geq\; \mathbb{E}[\mathcal{P}_{\bar{c},K,\epsilon}^{\mathrm{low}}] - O(\bar{c}).$$

Using $\mathbb{E}[\mathcal{P}_{\bar{c},K,\epsilon}^{\mathrm{low}}] \geq \bar{c}\ln(1/\bar{c}) - O(1/K) - O(\varepsilon)$ and $\bar{c} \asymp 1/\sqrt{N}$ yields $\Omega(\ln N/\sqrt{N})$. $\qquad\square$

---

[7] Shifting $\varepsilon$ mass within $[0,1]$ changes the expectation by at most $O(\varepsilon)$. The Riemann–sum error from replacing the continuous ERD by its $1/K$ grid is $O(1/K)$.

