# OpenReview forum: "Breaking Algorithmic Collusion in Human-AI Ecosystems"
_ICLR.cc/2026/Conference — Submitted to ICLR 2026_

### Official Review · Reviewer_vz5R · 2025-10-29

**Soundness:** 3
**Presentation:** 4
**Contribution:** 1
**Rating:** 2
**Confidence:** 3

**Summary:**

The paper studies collusion dynamics in pricing between different kinds of players. In particular, AI agents are modeled as colluding and human players are modeled as defecting from that collusions. The paper derives theoretical results that characterize the robustness of collision to defection under different numbers of players and types of strategic adaptation.

**Strengths:**

The paper is on an interesting topic and very well written and motivated. The theoretical results do speak to the authors research questions and as a non-theorist I found them interesting.

**Weaknesses:**

I don’t believe ICLR is the appropriate venue for this work. There is very little content here related to AI/ML except in the cover story of the variables. The works cited support this as most of the work is from economics, algorithmic game theory and related fields.

**Questions:**

Can we connect these theoretical results to empirical modeling with actual LLM or DeepRL agents?

Why do we need to assume that one player is the “human” and one is the “LLM”. Human’s might also collude. LLMs might not? It feels that this assumption is doing a lot of work and potentially confusing what the results actually mean.

---

> ### Author Response · Authors · 2025-11-26
> **Response to Reviewer vz5R**
>
> Thank you for your thoughtful review. We respond to your core points below.
>
> **Relevance to ICLR.** Our work fits within a stream of papers (e.g., [1], [2], [3]) studying strategic behavior and emergent multi-agent phenomena when AI agents, such as LLM agents and RL agents, repeatedly interact at test-time. We view our paper as using tools from game theory to shed light on this core problem in multi-agent AI ecosystems. Repeated pricing games serve as a clean testbed for evaluating collusion, and more broadly safety and robustness,  in multi-agent AI ecosystems.
>
> Our theoretical results complement recent empirical observations by showing that collusive behaviors collapse in mixed human–AI ecosystems with sufficiently many agents. We hope that our work inspires future empirical work to investigate how the number of agents affects the fragility of collusion.
>
> [1] Hammond et al. Multi-agent risks from advanced ai, 2025.
> [2] Fish, Yannai A. Gonczarowski, and Ran I. Shorrer. Algorithmic collusion by large language models, 2025.
> [3] Motwani et al., Secret collusion among ai agents: Multi-agent deception via steganography, 2025.
>
>
> **Modeling AI agents using repeated-game equilibrium strategies.** We first clarify that our results hold under much more relaxed assumptions on the strategies played by the AI agents in our model:
> All of our results extend directly to the weaker adoption equilibrium, which requires only that an AI agent receives more utility than a defection to a no-regret strategy (Section 4.1). This relaxed environment captures the “adoption” requirement that an AI agent outperforms a human.
> For multiple defectors, no equilibrium assumption at all is required: Theorem 3.6 holds for arbitrary strategies of the non-defecting players and arbitrary identities of defectors, with matching lower bounds.
> Our main results for the single-defector case (Section 3.2-3.3) already use a slightly milder Θ(1/T)-approximate equilibrium.
>
> We also clarify our motivation for modelling AI agents as an equilibrium strategy in the repeated game. As AI agents such as LLM agents (Appendix B.2) and RL agents (Appendix B.1) improve at optimizing over a long time horizon, we expect that these agents will become increasingly able to best respond to one another in algorithm space, not just in the underlying one-shot game. Empirical work in pricing and repeated games already shows AI systems implementing contingent, threat-like strategies. The solution concept of repeated-game equilibrium thus serves as a stylized model for the policy-level commitments of sophisticated agents.
>
> **Human vs. AI behavior.** No-regret dynamics are a standard model of human behaviors in auctions and pricing [4], capturing heuristic or boundedly rational behavior. We adopt this standard modelling choice in this work. This, coupled with our modelling assumption for AI agents, means that we focus on interactions between short-horizon/no-regret behavior and long-horizon strategic behavior to capture human-AI ecosystems.
>
> [4] Denis Nekipelov, Vasilis Syrgkanis, Eva Tardos.  Econometrics for Learning Agents Authors. EC 2015.
>
> **Human vs. AI collusion.** We build on a common perspective that AI  agents pose a greater threat of collusion than purely human ecosystems. This perspective comes from regulatory requirements which place clearer requirements on humans than algorithms or AI agents (e.g., [5]), and is also backed by empirical evidence in laboratory experiments (e.g., [6]). Our model of humans as no-regret dynamics captures this difference, since no-regret dynamics lead to lower levels of collusion than repeated game equilibrium [7].
>
> [5] Hartline, Long, Zhang. Regulation of Algorithmic Collusion, 2024.
> [6] Normann and Sternberg. Human-Algorithm Interaction: Algorithmic Pricing in Hybrid Laboratory Markets. 2022.
> [7] Feldman, Lucier, and Nisan. Correlated and coarse equilibria of single-item auctions. WINE 2016.

---

> > ### Comment · Reviewer_vz5R · 2025-11-28
> >
> > I appreciate the response. I will discuss relevance to ICLR with the other reviewers.

---

### Official Review · Reviewer_QZzb · 2025-10-29

**Soundness:** 3
**Presentation:** 3
**Contribution:** 3
**Rating:** 6
**Confidence:** 3

**Summary:**

This paper analyzes the stability of algorithmic collusion in iterated pricing games when one or more of the players "defects" against the others by following a no-regret strategy. Algorithmic collusion is an interesting setting of study, as it models the interactions between AI systems and humans in markets where collusion (e.g. high price fixing) can be detrimental for consumers. These kinds of scenarios are becoming extremely relevant as agentic AI is deployed in the wild.

The authors prove a few results in different settings:

**One defector:** If a single seller from the median-profit set defects (by switching to a follow-the-leader strategy), the market price falls to about $\frac{1+\log N}{N}$ where $N$ is the number of players.

**Many defectors:**  The authors show that adding more no-regret defectors drives prices down exponentially in M (the number of defectors).

**Defection-aware AI (extension):** If the AI agents can re-equilibrate after observing a defection, prices remain high: at least $1 - \frac{1}{K} - \frac{r(T)}{T}$ where $K$ is the number of discrete actions (product of discretizing the prices into K buckets), $T$ is the time horizon of the game and $r(T)$ is the external regret of the no-regret strategy (defection strategy).

**Strengths:**

This is a very well written paper that tackles the timely issue of algorithmic collusion. In particular the results, although theoretical, give practical, actionable insights that are useful to alleviate algorithmic collusion caused by non-myopic multi-agent interactions.

1. **Clarity:** The paper is extremely well written. All the necessary background is covered as well as important mathematical definitions required to analyze and interpret the results.

2. **Importance of the problem:** The problem of study is becoming increasingly relevant as agentic systems are deployed and forced to interact with humans and amongst each other. This kind of work can be very beneficial to prevent and mitigate potential negative consequences of multi-agent interaction; algorithmic collusion for price setting is an especially likely one to occur. More specifically, "manually" introducing defections (by setting low prices) is proven to drive the time-average of the prices down in iterated Bertrand games.

3. **Mathematical rigor:** Results are proved with tight (or matching) bounds in the single- and multi-defector regimes, plus clean extensions.

**Weaknesses:**

Although the paper is very rigorous it has some major weaknesses that reduce my confidence in accepting it:

1. **Lack of experimental results:** No experiments (even toy) to illustrate convergence of prices under standard online learners or simple scripted LLM agents. Empirical evidence could potentially ground the theoretical results. Authors also note this is a stylized model.

2. **Modeling assumptions:** The assumptions that most weaken the paper, in my view, are: Equilibrium/coordination: core results presume AI sellers are already in a stable repeated-game equilibrium (and, in the extension, can quickly re-equilibrate after a defection), which abstracts away the messy training and deployment dynamics that often prevent such coordination; Which seller defects: the sharpest guarantees rely on the defector coming from the median-profit set, a “typical seller” condition that may not hold in many markets and can materially change the predicted price drop if high-profit leaders defect instead.

3. **Relevance to ICLR:** Given that the paper has no experimental part and that the results come from explicit modeling of the interaction dynamics between agents, there are clearly no learning representations aspect in the paper (deep learning).

I would be willing to reconsider my score and be more confident in my assessment if the authors mitigate/address the issues mentioned above.

**Questions:**

1. How often is the “median-profit defector” condition expected in realistic markets? Can the bound extend to arbitrary defectors (e.g. lower-profit defectors)?
2.  How sensitive are the main results of the paper to different tie-breaking rules? Could the authors give their intuition?

---

> ### Author Response · Authors · 2025-11-26
> **Response to Reviewer QZzb**
>
> Thank you for your thoughtful review. We respond to your core points below.
>
>
> **Identity of player defecting.** We note that our results for multiple defectors do not require that the defectors are in the median profit set (Section 3.4). On the other hand, the example in Section 3.2  shows that such an assumption is necessary for the single defector case, in that a defection by a single higher-profit player may be insufficient to prevent collusion. This finding  suggests that regulators may benefit from auditing multiple firms or auditing a median-size firm, rather than only auditing the dominant player (L314-317).
>
> **Sensitivity to tie-breaking rules.** Our results are not sensitive to the choice of tiebreaking rule. In particular, our main results in Sections 3.2-3.4 all readily generalize to any tiebreaking rule. We stated the uniform-at-random tiebreaking rule for ease of exposition in the example in Section 3.1.
>
> **Relevance to ICLR.** Our work fits within a stream of papers (e.g., [1], [2], [3]) studying strategic behavior and emergent multi-agent phenomena when AI agents, such as LLM agents and RL agents, repeatedly interact at test-time. We view our paper as using tools from game theory to shed light on this core problem in multi-agent AI ecosystems. Repeated pricing games serve as a clean testbed for evaluating collusion, and more broadly safety and robustness,  in multi-agent AI ecosystems.
>
> Our theoretical results complement recent empirical observations by showing that collusive behaviors collapse in mixed human–AI ecosystems with sufficiently many agents. We hope that our work inspires future empirical work to investigate how the number of agents affects the fragility of collusion.
>
> [1] Hammond et al. Multi-agent risks from advanced ai, 2025.
> [2] Fish, Yannai A. Gonczarowski, and Ran I. Shorrer. Algorithmic collusion by large language models, 2025.
> [3] Motwani et al., Secret collusion among ai agents: Multi-agent deception via steganography, 2025.
>
> **Modeling AI agents using repeated-game equilibrium strategies.** We first clarify that our results hold under much more relaxed assumptions on the strategies played by the AI agents in our model:
> All of our results extend directly to the weaker adoption equilibrium, which requires only that an AI agent receives more utility than a defection to a no-regret strategy (Section 4.1). This relaxed environment captures the “adoption” requirement that an AI agent outperforms a human.
> For multiple defectors, no equilibrium assumption at all is required: Theorem 3.6 holds for arbitrary strategies of the non-defecting players and arbitrary identities of defectors, with matching lower bounds.
> Our main results for the single-defector case (Section 3.2-3.3) already use a slightly milder Θ(1/T)-approximate equilibrium.
>
> We also clarify our motivation for modelling AI agents as an equilibrium strategy in the repeated game. As AI agents such as LLM agents (Appendix B.2) and RL agents (Appendix B.1) improve at optimizing over a long time horizon, we expect that these agents will become increasingly able to best respond to one another in algorithm space, not just in the underlying one-shot game. Empirical work in pricing and repeated games already shows AI systems implementing contingent, threat-like strategies. The solution concept of repeated-game equilibrium thus serves as a stylized model for the policy-level commitments of sophisticated agents.

---

### Official Review · Reviewer_AyMA · 2025-11-01

**Soundness:** 2
**Presentation:** 2
**Contribution:** 2
**Rating:** 4
**Confidence:** 3

**Summary:**

This work analyzes algorithmic collusion in repeated $N$-player pricing game with a discrete price grid in mixed human-AI ecosystems. AI agents are assumed to play (approximate) equilibrium strategies in the repeated game, whereas human players are assumed to defect by switching to no-regret strategies. The paper analyzes how such defections affect the market price with a single (median-profit) defector, multiple defectors, greedy/FTL defectors, and defection-aware AI agents. This paper provides theoretical proofs on how algorithmic collusion varies under defections.

**Strengths:**

1. The overall writing is easy to follow. No major errors.
2. Precise formalization and clear proofs of theorems and lemmas.
3. Useful scope of extensions. Covers single vs. multiple defectors, no-regret vs. FTL, and defection-aware agents, which sharpens the paper’s scope.

**Weaknesses:**

1. No empirical evidence. This paper is purely theoretical without any experiments or simulations of RL agents or LLM agents. Since it claims relevance to RL/LLM agents, adding empirical evidence would validate the theory in practice.
2. This paper assumes RL/LLM agents would play equilibrium strategies. However, prior work suggests RL [1,2] and LLMs [3,4] may not be able to converge to Nash equilibria in repeated interactions and results vary across different RL algorithms and LLMs. Either justify this or soften the assumption.
3. Lack of convergence proof of AI algorithms under defections. There is no convergence/stability analysis for RL/LLM algorithms under defections (and no experiments). Provide a convergence claim for a representative algorithm class showing trained agents approximate the analyzed scenarios.

[1] Mazumdar, E., Ratliff, L. J., Jordan, M. I., & Sastry, S. S. (2019). Policy-gradient algorithms have no guarantees of convergence in linear quadratic games. _arXiv preprint arXiv:1907.03712_.

[2] Zhang, K., Yang, Z., & Başar, T. (2021). Multi-agent reinforcement learning: A selective overview of theories and algorithms. _Handbook of reinforcement learning and control_, 321-384.

[3] Akata, E., Schulz, L., Coda-Forno, J., Oh, S. J., Bethge, M., & Schulz, E. (2025). Playing repeated games with large language models. _Nature Human Behaviour_, 1-11.

[4] Lorè, N., & Heydari, B. (2024). Strategic behavior of large language models and the role of game structure versus contextual framing. _Scientific Reports_, _14_(1), 18490.

**Questions:**

1. What theoretical evidence supports the assumption that deployed AI (RL/LLM) agents play (approximate) equilibrium strategies in this setting?
2. Can you add empirical analysis with standard RL/LLM agents to support theoretical results?
3. Alternatively, can you provide a convergence/stability guarantee (or a clear non-convergence statement) for a representative RL algorithm under defections?
4. Could you include a brief comparison table listing the closest papers and, for each: assumptions, agent models, main results? Please highlight what is new here and note where your assumptions match or differ from prior work.

---

> ### Author Response · Authors · 2025-11-26
> **Response to Reviewer AyMA**
>
> Thank you for your thoughtful review. We respond to your core points below.
>
> **Relevance to AI agents.** Our work fits within a stream of papers (e.g., [1], [2], [3]) studying strategic behavior and emergent multi-agent phenomena when AI agents, such as LLM agents and RL agents, repeatedly interact at test-time. We view our paper as using tools from game theory to shed light on this core problem in multi-agent AI ecosystems. Repeated pricing games serve as a clean testbed for evaluating collusion, and more broadly safety and robustness,  in multi-agent AI ecosystems.
>
> Our theoretical results complement recent empirical observations by showing that collusive behaviors collapse in mixed human–AI ecosystems with sufficiently many agents. We hope that our work inspires future empirical work to investigate how the number of agents affects the fragility of collusion.
>
> [1] Hammond et al. Multi-agent risks from advanced ai, 2025.
> [2] Fish, Yannai A. Gonczarowski, and Ran I. Shorrer. Algorithmic collusion by large language models, 2025.
> [3] Motwani et al., Secret collusion among ai agents: Multi-agent deception via steganography, 2025.
>
> **Modeling AI agents using repeated-game equilibrium strategies.** We first clarify that our results hold under much more relaxed assumptions on the strategies played by the AI agents in our model:
> All of our results extend directly to the weaker adoption equilibrium, which requires only that an AI agent receives more utility than a defection to a no-regret strategy (Section 4.1). This relaxed environment captures the “adoption” requirement that an AI agent outperforms a human.
> For multiple defectors, no equilibrium assumption at all is required: Theorem 3.6 holds for arbitrary strategies of the non-defecting players and arbitrary identities of defectors, with matching lower bounds.
> Our main results for the single-defector case (Section 3.2-3.3) already use a slightly milder Θ(1/T)-approximate equilibrium.
>
> We also clarify our motivation for modelling AI agents as an equilibrium strategy in the repeated game. As AI agents such as LLM agents (Appendix B.2) and RL agents (Appendix B.1) improve at optimizing over a long time horizon, we expect that these agents will become increasingly able to best respond to one another in algorithm space, not just in the underlying one-shot game. Empirical work in pricing and repeated games already shows AI systems implementing contingent, threat-like strategies. The solution concept of repeated-game equilibrium thus serves as a stylized model for the policy-level commitments of sophisticated agents. We agree that this assumption is simplifying and abstracts away many of the convergence issues noted by the reviewer.

---

### Official Review · Reviewer_2vej · 2025-11-01

**Soundness:** 3
**Presentation:** 2
**Contribution:** 2
**Rating:** 4
**Confidence:** 3

**Summary:**

The paper investigates how human defections from algorithmic collusion influence prices in mixed human–AI markets. They model human defections as no-regret or follow-the-leader (FTL) strategies, AI collusion as prices sustained by equilibrium strategies and human-AI markets as repeated Bertrand pricing games. Their analysis shows that when a typical agent (with median profit) switches from an equilibrium strategy (AI) to a no-regret strategy (human), the market price goes down by a polynomial factor of the number of players and exponentially when multiple humans defect, regardless of their identities. They also show that equilibrium collusion can persist when AI agents are defection-aware. All claims are proven analytically, with tight matching bounds and detailed proofs in the appendix.

**Strengths:**

The paper presents a stylised setting to analytically characterise the price effects caused by agents deviating from the equilibrium profile and makes a solid theoretical contribution. The methodology appears to be correct and reproducible. Generally, the paper is well structured with clarity in the definitions and the whole approach. The problem the authors are investigating is timely and needs research attention.

**Weaknesses:**

The authors do acknowledge the limitations of the simplifications in the current work for analytical tractability. However, I find that the motivation that AI agents play Nash equilibria in the repeated game needs strengthening. The authors use the terminology of “AI agents” and “no-regret learners,” but mathematically treat them as given strategy classes, not as adaptive processes. Thus, while the topic (AI collusion) is socially and technically relevant, the learning connection is mainly in the narrative of the paper. So the core contribution is a game-theoretic characterisation of equilibrium outcomes under mixed strategy profiles, not an analysis of learning dynamics or convergence.

**Questions:**

The approach of the authors and their theoretical contributions is a stepping stone for more research in the field. I believe the paper would be strengthened a lot if they provided at least some simulations with simple learning agents to better persuade the validity of their assumptions for the AI agents and show how their theory indeed bridges empirical observations. Also, I found the modelling rather orthogonal to the standard AI modeling, e.g., the Nisan and Kolumbus (2022), where AI are algorithms play the game repeatedly using some algorithm and agents either report their values in a meta-game or play a Nash equilibrium (or something else less sophisticated). Is there a reason that this paper models the human as playing the algorithm and the AI as playing the Nash equilibrium? From this, I don't really understand why something is termed "AI" and something is "human" - also, I don't understand why something should be termed algorithmic collusion when this is not the result of some algorithms training together and reaching a collusive outcome, but rather an assumption that a Nash equilibrium is played. In general, I don't see why something is distinctively AI vs human here and maybe the authors could clarify that.

---

> ### Author Response · Authors · 2025-11-26
> **Response to Reviewer 2vej**
>
> Thank you for your thoughtful review. We respond to your core points below.
>
> **Relevance to AI agents.** Our work fits within a stream of papers (e.g., [1], [2], [3]) studying strategic behavior and emergent multi-agent phenomena when AI agents, such as LLM agents and RL agents, repeatedly interact at test-time. We view our paper as using tools from game theory to shed light on this core problem in multi-agent AI ecosystems. Repeated pricing games serve as a simplified testbed for evaluating collusion, and more broadly safety and robustness, in multi-agent AI ecosystems.
>
> Our theoretical results complement recent empirical observations by showing that collusive behaviors collapse in mixed human–AI ecosystems with sufficiently many agents. We hope that our work inspires future empirical work to investigate how the number of agents affects the fragility of collusion.
>
> [1] Hammond et al. Multi-agent risks from advanced ai, 2025.
> [2] Fish, Yannai A. Gonczarowski, and Ran I. Shorrer. Algorithmic collusion by large language models, 2025.
> [3] Motwani et al., Secret collusion among ai agents: Multi-agent deception via steganography, 2025.
>
> **Modeling AI agents using repeated-game equilibrium strategies.** We first clarify that our results hold under much more relaxed assumptions on the strategies played by the AI agents in our model:
> All of our results extend directly to the weaker adoption equilibrium, which requires only that an AI agent receives more utility than a defection to a no-regret strategy (Section 4.1). This relaxed environment captures the “adoption” requirement that an AI agent outperforms a human.
> For multiple defectors, no equilibrium assumption at all is required: Theorem 3.6 holds for arbitrary strategies of the non-defecting players and arbitrary identities of defectors, with matching lower bounds.
> Our main results for the single-defector case (Section 3.2-3.3) already use a slightly milder Θ(1/T)-approximate equilibrium.
>
> We also clarify our motivation for modelling AI agents as an equilibrium strategy in the repeated game. As AI agents such as LLM agents (Appendix B.2) and RL agents (Appendix B.1) improve at optimizing over a long time horizon, we expect that these agents will become increasingly able to best respond to one another in algorithm space, not just in the underlying one-shot game. Empirical work in pricing and repeated games already shows AI systems implementing contingent, threat-like strategies. The solution concept of repeated-game equilibrium thus serves as a stylized model for the policy-level commitments of sophisticated agents.
>
>
> **Human vs. AI behavior.** No-regret dynamics are a standard model of human behaviors in auctions and pricing [4], capturing heuristic or boundedly rational behavior. We adopt this standard modelling choice in this work. This, coupled with our modelling assumption for AI agents, means that we focus on interactions between short-horizon/no-regret behavior and long-horizon strategic behavior to capture human-AI ecosystems.
>
> [4] Denis Nekipelov, Vasilis Syrgkanis, Eva Tardos.  Econometrics for Learning Agents Authors. EC 2015.
>
> **Why call this “algorithmic collusion”?** This question gets to the heart of ongoing debate in the literature (e.g., [5], [6]). As pricing is increasingly delegated to algorithms, strategic behavior occurs in algorithmic space, where repeated-game best responses can sustain supracompetitive prices—even though one-shot Nash would not. Thus purely “rational,” non-communicating algorithms can generate outcomes harmful to consumers, without explicit coordination or shared information. Many works (e.g., [7], [8]) refer to such outcomes as algorithmic collusion, because the market consequences mirror classical collusion. Whether one prefers the term or not, the phenomenon—sustained supracompetitive pricing due to algorithmic strategic behavior—is real, well-documented, and precisely what we characterize.
>
> [5] Hartline, Long, Zhang. Regulation of Algorithmic Collusion, 2024.
> [6] Arunachaleswaran, Collina, Kannan, Roth, and Ziani. Algorithmic Collusion Without Threats. ITCS 2025.
> [7]  Calvano, Calzolari, Denicolo, Pastorello. Artificial intelligence, algorithmic pricing, and collusion. American Economic Review, 110(10):3267–3297, 2020.
> [8] Brown and MacKay. Competition in pricing algorithms. American Economic Journal: Microeconomics, 15(2):109–156, 2023.

---

> > ### Comment · Reviewer_2vej · 2025-11-28
> > **Acknowledgement of rebuttal**
> >
> > Thank you for the rebuttal. I acknowledge your response and have no further questions at this stage. I will discuss the evaluation with the other reviewers.

---

### Meta-Review · Area_Chair_uqQy · 2026-01-18

**Summary:**

The paper theoretically examines repeated pricing games in mixed ecosystems of AI agents and humans, demonstrating that algorithmic collusion is generally fragile when even a single human defects to a simple no-regret strategy. The authors find that while a single defector can drive the market prices down and multiple defectors push prices close to competitive levels, high prices can still persist if the AI agents are "defection-aware" and dynamically adapt to the human's behavior.

**Reviewer Concerns:**

The main concerns of the reviewers were the lack of empirical evaluation and was also pointed out that the paper may not be appropriate for this venue and criticised the framework/model. The AC liked the theoretical work of the paper but has to agree with the criticism of the reviewers.

**Reviewer Scores:**

One reviewer was slightly positive and the rest were negative or slightly negative. We believe that the authors gave a solid rebuttal and there is a possibility that the reviewers would have considered raising their scores. Nevertheless, the AC does not feel comfortable enough to accept the paper given the current scores, we believe the paper is a borderline paper and for calibration reasons we recommend weak rejection.

---

### Decision · Program_Chairs · 2026-01-26

Reject